# ToaSt: Token Channel Selection and Structured Pruning for Efficient ViT

**Hyunchan Moon** [1]   **Cheonjun Park** [2]   **Steven L. Waslander** [3]

## Abstract

Vision Transformers (ViTs) have achieved remarkable success across various vision tasks, yet their deployment is often hindered by prohibitive computational costs. While structured weight pruning and token compression have emerged as promising solutions, they suffer from prolonged retraining and inter-layer dependencies that complicate optimization, respectively. We propose ToaSt, a decoupled framework applying specialized strategies to distinct ViT components. We apply coupled head-wise structured pruning to Multi-Head Self-Attention modules, leveraging attention operation characteristics to enhance robustness. For Feed-Forward Networks (over 60% of FLOPs), we introduce Token Channel Selection (TCS), a training-free method that filters redundant noise channels at inference time. Extensive evaluations across nine diverse models, including DeiT, ViT-MAE, and Swin Transformer, demonstrate that ToaSt achieves superior trade-offs between accuracy and efficiency, consistently outperforming existing baselines. On ViT-MAE-Huge, ToaSt achieves 88.52% accuracy (+1.64%p) with 39.4% FLOPs reduction. ToaSt also transfers effectively to diverse downstream tasks (COCO detection, ADE20K segmentation, CIFAR-100 classification), achieving 52.2 versus 51.9 mAP on COCO. Code: github.com/SHANNonLab-HUFS/ToaSt.

## 1. Introduction

Vision Transformers (ViTs) (Dosovitskiy, 2020) have achieved remarkable success across a wide range of computer vision tasks, including image classification (Touvron et al., 2021), object detection (Liu et al., 2021), and semantic segmentation. By leveraging self-attention mechanisms to

[1]LG Electronics, Seoul, Republic of Korea [2]Hankuk University of Foreign Studies, Yongin, Republic of Korea [3]University of Toronto, Toronto, Canada. Correspondence to: Cheonjun Park <jun@hufs.ac.kr>.

*Proceedings of the 43rd International Conference on Machine Learning*, Seoul, South Korea. PMLR 306, 2026. Copyright 2026 by the author(s).

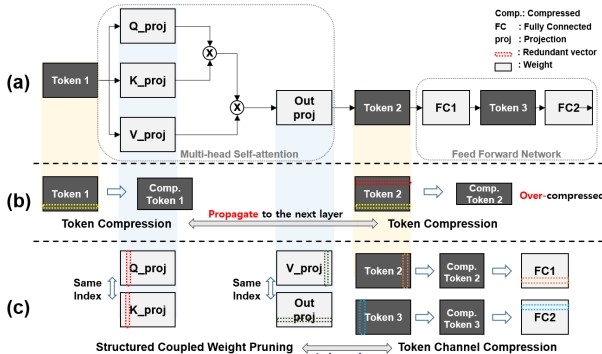

*Figure 1.* **ToaSt compression methodology.** (a) Standard ViT block architecture. (b) Token compression propagates compression effects across layers due to inter-layer dependencies. (c) ToaSt independently compresses each layer through coupled weight pruning (MHSA) and token channel selection (FFN), preventing cross-layer propagation while reducing $d_k$ and $D$ dimensions.

capture global dependencies, ViTs have demonstrated competitive or superior performance compared to convolutional neural networks, and have become foundational architectures for multimodal learning (Radford et al., 2021; Li et al., 2022a). However, this representational power comes at a significant computational cost: ViTs are substantially heavier than CNNs of comparable accuracy, and their performance advantages only emerge at larger scales of both data and network size, presenting critical challenges for deployment in resource-constrained environments such as mobile devices and edge computing platforms.

The cost stems from two sources: the quadratic self-attention complexity $\mathcal{O}(N^2)$ in sequence length $N$, and the linear projections in Multi-Head Self-Attention (MHSA) and the Feed-Forward Network (FFN) that scale with the hidden dimension $D$. In standard ViTs, FFN layers dominate the computational cost, contributing approximately 61% of total FLOPs, while MHSA accounts for less than 40% (Kong et al., 2022) (Table 6). To address these costs, **weight pruning methods** (Chen et al., 2021; Yu et al., 2022) structurally remove channels, heads, or blocks, but rely on costly full-model retraining and are not specifically designed to exploit the dominant FFN redundancy. **Token compression methods** (Rao et al., 2021; Liang et al., 2022; Bolya et al., 2022; Chen et al., 2023) reduce the sequence length $N$, directly targeting attention's quadratic cost; however, they operate exclusively on the sequence dimension and cannot

address the dominant $D^2$ complexity in FFNs. Furthermore, token-level decisions propagate globally to subsequent layers, creating inter-layer dependencies that complicate the optimization landscape.

We propose **ToaSt**, a decoupled framework that eliminates the retraining overhead of weight pruning while targeting the FFN channel redundancy that token compression cannot address, through two complementary components:

**Structured Coupled Weight Pruning for MHSA.** We reduce the per-head dimension $d_k$ by pruning corresponding indices across coupled weight matrices ($\mathbf{W}_Q$, $\mathbf{W}_K$, $\mathbf{W}_V$, $\mathbf{W}_{\text{proj}}$) while preserving the block interface, enabling layer-independent compression.

**Token Channel Selection (TCS) for FFN.** Through empirical analysis of FFN activations, we identify characteristic redundancy signatures in deeper layers—high sparsity, low effective rank, and high $R^2$ reconstruction fidelity. TCS exploits these signatures via a training-free, layer-adaptive strategy that dynamically selects channels per FFN sub-layer (FC1, FC2), directly reducing the $D^2$ complexity without retraining. Notably, TCS acts as an implicit noise filter, yielding consistent accuracy gains across architectures.

On ImageNet-1K (Deng et al., 2009), ToaSt achieves **88.52% Top-1 on ViT-MAE-Huge** (+1.64%p) with 39.4% FLOPs reduction; recovery requires only ∼15 fine-tuning epochs for ViT-MAE-Huge versus ∼290 for DeiT-Base, indicating that larger models benefit disproportionately from our approach. ToaSt also transfers to downstream tasks, reaching **52.2 mAP** on COCO (Cascade R-CNN, Swin-Base) versus the 51.9 mAP baseline.

Our contributions are summarized as follows:

- We present a structured coupled weight pruning method for MHSA that reduces per-head dimensions by pruning corresponding indices across coupled weight matrices (Q-K, V-Proj pairs), enabling layer-independent compression (Section 3.1).
- We provide empirical analysis of FFN activation patterns revealing redundancy signatures in deeper layers, and introduce Token Channel Selection (TCS), a training-free approach with layer-adaptive ratios that filters redundant noise and eliminates retraining overhead (Section 3.2).
- We demonstrate that ToaSt achieves superior accuracy-efficiency trade-offs across nine ViT models (DeiT, ViT-MAE, Swin) and transfers to downstream object detection, with larger models requiring fewer fine-tuning epochs (Section 4).

## 2. Related Works

**Token Compression** Reducing sequence length ($N$) is the standard approach to mitigate quadratic attention costs.

Early *pruning* methods including EViT (Liang et al., 2022) and DynamicViT (Rao et al., 2021) discard less informative tokens. To prevent information loss from hard removal, *token merging* strategies have emerged: ToMe (Bolya et al., 2022) aggregates similar tokens via bipartite matching, while DiffRate (Chen et al., 2023) learns differentiable compression rates. Recently, PiToMe (Tran et al., 2024) advances this by leveraging spectral clustering to protect informative tokens before merging. These concepts have also expanded to LLMs and VLMs for efficient KV-cache management (Zhang et al., 2023; Li et al., 2024) and visual token reduction (Chen et al., 2024). However, these methods reduce FFN computation only linearly with $N$, leaving the dominant hidden-dimension complexity $\mathcal{O}(D^2)$ unaddressed. Our ToaSt complements these sequence-level reductions by directly targeting the orthogonal channel dimension in FFNs.

**Structured Pruning.** Structured pruning reduces model complexity by removing coherent parameter groups (heads, channels), enabling direct acceleration on standard GPUs without specialized sparse kernels. Due to this practicality, various methods have been proposed to identify unimportant structures in ViTs, often utilizing magnitude-based or gradient-based criteria (Yang et al., 2021; Fang et al., 2023; Yu et al., 2022). However, broadly removing entire structural components typically leads to significant accuracy drops, necessitating expensive and time-consuming retraining (fine-tuning) to recover performance. This heavy computational overhead limits their applicability to large-scale foundation models. To address this limitation, our ToaSt employs activation-based statistics for structured removal in ViTs, achieving significant compression without the heavy retraining costs of global pruning methods.

**Joint and Hybrid Methods.** Hybrid strategies aim to simultaneously optimize multiple dimensions. Recent frameworks integrate token compression with channel pruning (Wang et al., 2025; 2022) or combine pruning with quantization (Dong et al., 2023; Yuan et al., 2022) to maximize speedup. While effective, these strategies face distinct limitations: joint pruning methods often involve complex coupled optimization landscapes, whereas quantization-integrated approaches face hardware-dependent trade-offs—while INT8 and INT16 quantization runs efficiently on commodity hardware, ultra-low-bit methods ($\leq$4-bit) often rely on specialized hardware accelerators (Park et al., 2025b; Zou et al., 2025), custom GPU kernels (Park et al., 2024b; Ma et al., 2024), or hardware-aware codebook designs constrained by GPU cache hierarchies (Tseng et al., 2024) to achieve practical speedups. In contrast, our ToaSt decouples the compression problem: we employ structured coupled weight pruning for MHSA heads and training-free token channel selection for FFNs—both targeting the channel dimension while leaving the token sequence intact. This

separation simplifies the optimization landscape, allowing for aggressive compression with minimal recovery overhead.

## 3. ToaSt

We propose **ToaSt**, a compression framework built upon the philosophy of Layer-Independent Compression. ToaSt operates in two decoupled stages: (1) **MHSA Compression** (§3.1), which reduces the internal head dimension $d_k$ via structured coupled weight pruning, and (2) **FFN Compression** (§3.2), which mitigates depth-wise redundancy through analysis-driven channel selection. Algorithm 1 in Appendix B provides a unified pseudocode summary of the full pipeline.

**Prerequisites** Consider a Vision Transformer with $L$ layers. Let $\mathbf{X} \in \mathbb{R}^{N \times D}$ denote the input feature, where $N$ is the number of tokens and $D$ is the embedding dimension. For MHSA with $H$ heads, each head $h$ has internal dimension $d_k = D/H$ with weight matrices $\mathbf{W}_Q^h, \mathbf{W}_K^h, \mathbf{W}_V^h \in \mathbb{R}^{D \times d_k}$ and $\mathbf{W}_{\text{proj}}^h \in \mathbb{R}^{d_k \times D}$. For structured pruning, we use $\mathbf{W}_{QK}^h$ and $\mathbf{W}_{VO}^h$ to denote the jointly-considered weight pairs $(\mathbf{W}_Q^h, \mathbf{W}_K^h)$ and $(\mathbf{W}_V^h, \mathbf{W}_{\text{proj}}^h)$, respectively, used in coupled importance scoring. For FFN, $\mathbf{W}_{\text{FC1}} \in \mathbb{R}^{D \times 4D}$ and $\mathbf{W}_{\text{FC2}} \in \mathbb{R}^{4D \times D}$ denote the expansion and reduction projections. We use GM($\cdot$) for the geometric median (He et al., 2019) and primed notation (e.g., $d_k'$) for compressed dimensions.

### 3.1. Structured Coupled Weight Pruning for MHSA

To realize our layer-independent philosophy in the attention mechanism, we target the internal head dimension $d_k$ rather than the global embedding dimension $D$. This ensures full compatibility with residual connections and preserves the feature landscape for downstream layers.

**Step.1 Coupled Matrix Formulation and Constraints**
The MHSA module relies on coupled linear transformations. For a specific attention head $h$, the operations are defined as:

$$\mathbf{Q}^h = \mathbf{X}\mathbf{W}_Q^h, \quad \mathbf{K}^h = \mathbf{X}\mathbf{W}_K^h,$$
$$\mathbf{A}^h = \text{softmax}\left(\frac{\mathbf{Q}^h(\mathbf{K}^h)^\top}{\sqrt{d_k}}\right) \tag{1}$$
$$\mathbf{O}^h = \mathbf{A}^h\mathbf{V}^h\mathbf{W}_{\text{proj}}^h = \mathbf{A}^h(\mathbf{X}\mathbf{W}_V^h)\mathbf{W}_{\text{proj}}^h \tag{2}$$

where $\mathbf{X} \in \mathbb{R}^{N \times D}$ is the input. To maintain mathematical integrity during pruning, strict synchronization is required (Figure 2):

- **Q-K Synchronization:** Pruning column $j$ of $\mathbf{W}_Q^h$ necessitates pruning column $j$ of $\mathbf{W}_K^h$ to preserve the

dot-product validity.

- **V-Proj Synchronization:** Pruning column $j$ of $\mathbf{W}_V^h$ requires removing row $j$ of $\mathbf{W}_{\text{proj}}^h$ to maintain the output projection's inner dimension.

As empirically validated in Figure 3, ignoring these constraints (non-align) leads to a catastrophic accuracy collapse, whereas our coupled approach maintains functional integrity even at high pruning ratios.

**Step.2 Selection Criterion: Geometric Median-Based Importance** We employ a static importance criterion based on the **Geometric Median (GM)** (He et al., 2019) (Appendix C) of the pre-trained weights. The GM identifies redundant dimensions that are most *replaceable* by others within the same layer. Specifically, dimensions located closest to the weight distribution's center are considered to possess the highest redundancy, as their information can be most effectively approximated by the remaining dimensions.

Based on the synchronization constraints defined above, the unified importance score $I^h[j]$ for the $j$-th dimension of head $h$ is calculated as the Euclidean distance from the geometric median:

$$I_{QK}^h[j] = \left\|\mathbf{w}_{QK,j}^h - \text{GM}(\mathbf{W}_{QK}^h)\right\|_2$$
$$I_{VO}^h[j] = \left\|\mathbf{w}_{VO,j}^h - \text{GM}(\mathbf{W}_{VO}^h)\right\|_2 \tag{3}$$

where $\mathbf{w}_j^h$ denotes the $j$-th weight vector of the corresponding coupled matrix. Dimensions with the lowest importance scores are prioritized for pruning. We compare GM against $L_1$ and $L_2$ norm-based metrics, with results demonstrating GM's superior effectiveness in Appendix C.

**Step.3 Head-wise Uniform Strategy and Efficiency** We apply the scores from Eq. (3) using a *Head-wise Uniform Pruning* strategy, enforcing $d_k'(h) = d_k'$ for all $h$. This ensures computational regularity, enabling efficient batched matrix multiplication on standard hardware without padding overhead.

Following a layer-adaptive schedule—skipping the first layer and applying 90% pruning to the rest—we achieve a 90% reduction in MHSA FLOPs (Appendix Table 9).

**Fine-tuning Efficiency on Large Models.** We observe an inverse relationship between model scale and fine-tuning requirements: ViT-MAE-Huge recovers beyond baseline performance in only 15 epochs ($\sim$15 hours on 4$\times$ H100), whereas DeiT-Small requires 290 epochs ($\sim$1 day). This is substantially cheaper than conventional pruning methods that require full retraining (e.g., 300 epochs for DeiT). Detailed fine-tuning costs are provided in Appendix Table 10.

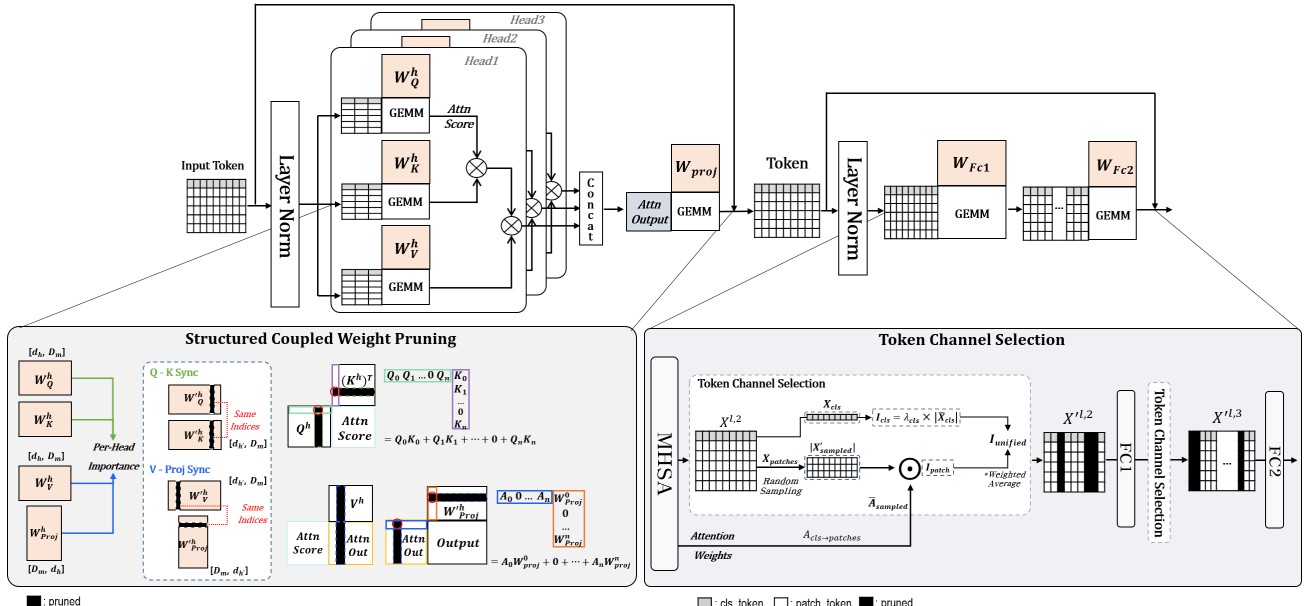

*Figure 2.* **Overview of the ToaSt framework for layer-independent compression. (a) Structured Coupled MHSA Weight Pruning:** Pruning indices are synchronized across coupled groups (Q-K and V-Proj) to reduce the internal head dimension $d_k$ while preserving the attention mechanism's functional integrity. **(b) Token Channel Selection for FFN:** Redundant channels in the intermediate FFN layer are identified and eliminated based on feature importance analysis (Eq. 6), maintaining the global embedding dimension $D$ at the block interface.

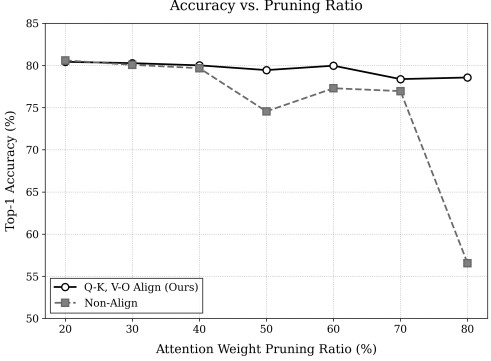

*Figure 3.* **Impact of coupled index synchronization on accuracy.** Under identical pruning ratios and the geometric median metric, Non-Align prunes Q/K/V/O with independent indices per matrix, while Ours (Align) enforces shared indices within Q-K and V-O pairs, significantly mitigating accuracy drop at high pruning ratios.

### 3.2. Token Channel Selection for FFN

While MHSA pruning targets the internal head dimension, the FFN module introduces significant redundancy through its channel expansion ($D \rightarrow 4D$).

As depicted in Figure 2(b), ToaSt addresses this by implementing an **Attention-Guided Token Channel Selection** mechanism. This method dynamically evaluates and prunes channels in the *expanded dimension* based on an empirical analysis of **Sparsity**, **Linear Reconstruction Fidelity** ($R^2$), and **Effective Rank**, effectively reducing the computational cost of both expansion (FC1) and reduction (FC2)

projections without altering the block interface.

#### 3.2.1. EMPIRICAL ANALYSIS OF FFN REDUNDANCY

We investigated the internal feature characteristics of pretrained ViTs (e.g., Swin-Base) to justify our design choices. As shown in Figure 4, we observe three critical phenomena that motivate our sampling-based pruning strategy:

**1) High Linear Reconstruction Fidelity ($R^2$).**

To validate the feasibility of partial channel observation, we measure the coefficient of determination ($R^2$) by reconstructing a specific channel's activations from a linear combination of others. For a target channel vector $\mathbf{y} \in \mathbb{R}^n$ (where $n$ is the number of tokens), we compute its linear reconstruction $\hat{\mathbf{y}}$ using the remaining channels in the same layer via least-squares optimization. The fidelity is defined as:

$$R^2 = 1 - \frac{\sum_{i=1}^{n}(y_i - \hat{y}_i)^2}{\sum_{i=1}^{n}(y_i - \bar{y})^2} \quad (4)$$

Empirical results show that the $R^2$ score consistently exceeds **0.9** across most layers (Figure 4, Center). This evidence demonstrates that the high-dimensional channels are linearly dependent, meaning the global importance distribution can be accurately estimated from a tiny subset of channels.

**2) Collapsing Effective Rank.**

We analyze the spectral properties of the feature matrix $\mathbf{X}$

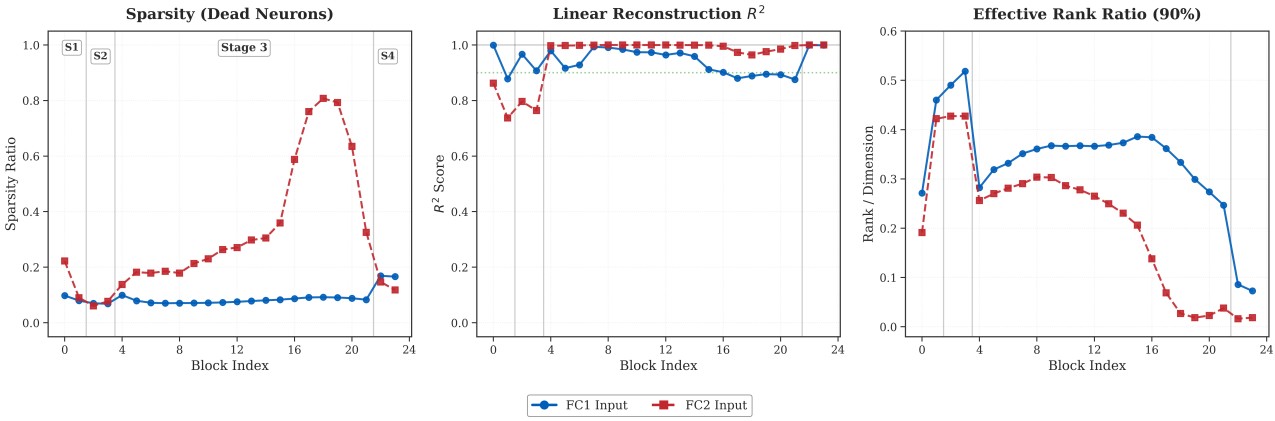

*Figure 4.* **Layer-wise redundancy analysis of Swin-Base FFN.** (Left) **Sparsity** increases in deeper stages, indicating many "dead neurons." (Center) **Linear Reconstruction** $R^2$ remains near 1.0, proving that feature channels are highly dependent. (Right) **Effective Rank Ratio:** the fraction of singular values needed to capture 90% of the total variance ($\min_k k/C$ s.t. $\sum_{i=1}^{k} \sigma_i^2 \geq 0.9 \cdot \sum_{j=1}^{C} \sigma_j^2$). A lower ratio indicates higher redundancy. The ratio collapses in later stages, confirming that the $4D$ expansion contains massive redundancy.

using the **Effective Rank Ratio**. This metric quantifies the data's intrinsic dimensionality:

$$\text{Rank}_{eff} = \frac{\exp(H(\bar{\boldsymbol{\sigma}}))}{C}, \quad \bar{\sigma}_i = \frac{\sigma_i}{\sum_{j=1}^{C} \sigma_j} \quad (5)$$

where $\sigma_i$ denotes the $i$-th singular value of the feature matrix $\mathbf{X}$, and $H(\bar{\boldsymbol{\sigma}}) = -\sum_{i=1}^{C} \bar{\sigma}_i \ln(\bar{\sigma}_i)$ denotes Shannon entropy. In practice, we measure redundancy using a PCA-based effective rank ratio: the minimum fraction of singular values $k/C$ whose cumulative squared sum captures 90% of the total variance. As shown in Figure 4 (Right), the effective rank exhibits a significant *collapse* in deeper layers.

**3) Increase in Sparsity.** As shown in Figure 4 (Left), the sparsity ratio (the proportion of activations satisfying $|x_c| < 0.1 \cdot \overline{|x|}$, where $x_c$ denotes the activation at channel $c$ and $\overline{|x|}$ is the mean absolute activation) increases significantly in later blocks. This phenomenon is driven by the GELU activation function, and as noted in recent studies, Transformers naturally exhibit higher activation sparsity in deeper layers (Li et al., 2022b). This trend suggests that a large portion of neurons contribute minimally to the final representation.

**4) Channel SNR Gap.** To further validate that pruned channels are indeed noise-dominant, we measure the signal-to-noise ratio (SNR) of kept vs. pruned channels in FC2-pruned blocks of Swin-Base. Across all pruned layers, kept channels exhibit 3–5.5× higher SNR than pruned channels (Appendix Table 8), confirming that TCS selectively removes low-discriminative channels that act as noise amplifiers rather than useful feature carriers. This provides a quantitative explanation for the accuracy improvement observed after pruning.

### 3.2.2. PROCESS: TOKEN CHANNEL SELECTION

**Step.1 Statistical Sampling for Efficient Thresholding** Based on the high $R^2$ observation, we introduce a *Training-free Statistical Sampling* strategy. Computing importance scores requires aggregating feature magnitudes across all $N$ tokens. This operation is computationally expensive ($O(N \cdot C)$), as it must be performed for both the embedding dimension ($C = D$) at the FC1 input and the expanded dimension ($C = 4D$) at the FC2 input.

To mitigate this, we estimate the channel importance using only a randomly sampled subset of tokens $\mathcal{S} \subset \{1, ..., N\}$. The sampling rate is adaptively determined based on the layer depth, ranging from **2% to 20%** of $N$ ($|\mathcal{S}| \in [0.02N, 0.2N]$). The strong linear dependency among channels ($R^2 \approx 1.0$) guarantees that this minimal subset is sufficient to accurately estimate the global importance distribution for any channel dimension $C$, reducing the analysis overhead by orders of magnitude.

**Step.2 Importance-Based Channel Selection** Using the sampled tokens, we compute an importance score $I_c$ for channel $c$. The formulation is architecture-dependent:

**(a) CLS-distilled models** (e.g., DeiT), where the CLS token is explicitly trained to encode class-discriminative features:

$$I_c = \lambda_{cls} |x_{cls}^{(c)}| + \lambda_{patch} \frac{1}{|\mathcal{S}|} \sum_{i \in \mathcal{S}} \left( A_{cls,i} \cdot |x_i^{(c)}| \right) \quad (6)$$

where $x_{cls}^{(c)}$ is the CLS token activation at channel $c$, $A_{cls,i}$ is the attention weight from CLS to patch $i$, and we set $\lambda_{cls} = 2.0, \lambda_{patch} = 1.0$ to prioritize channels encoding global semantic information.

**(b) Other architectures** (e.g., ViT-MAE, Swin Trans-

former), where no CLS-aligned training signal exists, the formulation simplifies to:

$$I_c = \frac{1}{|\mathcal{S}|} \sum_{i \in \mathcal{S}} |x_i^{(c)}| \qquad (7)$$

This reduces to a magnitude-based selection over sampled patch tokens. Ablation results (Appendix Table 16) confirm that attention-guided weighting benefits CLS-distilled models (+2.2 to +8.0%p) but is neutral for MAE and Swin, validating this architecture-dependent design.

**Hardware-Friendly Structured Reduction.** Crucially, this metric determines the survival of the entire channel index $c$. For each FFN sub-layer (FC1 and FC2 independently), TCS removes low-importance channels from the input tokens and slices the corresponding weight entries along the input dimension (i.e., removing the $c$-th column of the weight matrix), forming a dense sub-matrix for efficient GEMM. This structural regularity allows for **immediate acceleration on standard hardware (GPUs)** without the need for specialized sparse libraries or indexing overheads, following the GPU-friendly pruning paradigm (Park et al., 2024a; 2025a).

**Step.3 Layer-Adaptive Pruning Policy** Finally, we apply a differentiated pruning schedule aligned with our sparsity and rank analysis.

**FC1 (Expansion):** In early layers where Effective Rank is relatively high, we apply conservative pruning to preserve feature diversity.

**FC2 (Reduction):** In deeper layers where Sparsity is high and Rank is low, we aggressively prune (up to 90%) the redundant dimensions identified by Eq. (6), directly translating the "Collapsing Rank" observation into computational savings.

## 4. Experiments

### 4.1. Experimental Setup

**Models and Datasets.** We evaluate ToaSt on nine models across three families: DeiT (T/S/B) (Touvron et al., 2021), ViT-MAE (B/L/H) (He et al., 2022), and Swin Transformer (T/S/B) (Liu et al., 2021). Evaluations are conducted on ImageNet-1K (Deng et al., 2009) for classification (224 × 224), COCO 2017 (Lin et al., 2014) for object detection via Cascade Mask R-CNN (Cai & Vasconcelos, 2019) (multi-scale training, short side 480–800, long side ≤1333), and ADE20K for semantic segmentation (512 × 512). Throughput and latency are measured on a single **NVIDIA H100 GPU** (batch size 128, fp32).

**MHSA Structured Pruning (Fine-tuning):** We apply 90% channel reduction (80% for tiny models) to all layers except the first to preserve the initial embedding interface. The compressed weights are fine-tuned using AdamW with a cosine learning rate scheduler.

**FFN Token Channel Selection (Training-free):** We apply asymmetric pruning ratios at inference time without retraining. Based on redundancy analysis, we use conservative ratios (0–30%) for FC1 and aggressive ratios (50–90%) for FC2 in deeper layers. Importance is determined dynamically via the sampling-based statistical strategy described in Section 3.2. Specific per-layer pruning ratios are detailed in Appendix E.

### 4.2. ImageNet-1K Classification Results

Table 1 presents comprehensive comparison of ToaSt against state-of-the-art token compression methods including ToMe (Bolya et al., 2022) and DiffRate (Chen et al., 2023) across all evaluated architectures. We report Top-1/Top-5 accuracy, GFLOPs, FLOPs reduction ratio, and measured throughput on H100 GPU.

**Accuracy Improvement via Regularization.** ToaSt consistently improves accuracy over unpruned baselines despite significant FLOPs reduction. Specifically, DeiT-Base and ViT-MAE-Huge achieve gains of **+3.02%p** and **+1.64%p** with approximately 39% fewer FLOPs. This suggests that our token channel selection acts as an implicit regularizer, effectively removing redundant noise and enhancing generalization.

**Superior Efficiency Trade-offs.** Compared to token compression methods (Table 2), ToaSt provides a better accuracy-efficiency trade-off. By targeting channel dimensions instead of sequence length, our approach maintains higher representational density, yielding superior speedups and accuracy at equivalent FLOPs budgets.

**Hardware-Level Acceleration.** ToaSt translates theoretical FLOPs reduction into substantial hardware throughput gains on the NVIDIA H100. Throughput improves by **1.28× to 2.07×** across architectures. Notably, DeiT-Small achieves 4783.3 img/s (2.07× speedup) while improving accuracy by +3.58%p, validating the hardware-friendliness of structured pruning.

The disproportionately large speedups on smaller DeiT models (over 2×) are explained by two factors. First, MHSA consumes a larger relative share of total latency in smaller models, so aggressive head-dimension pruning (80% for DeiT-Tiny, 90% for DeiT-Small/Base) yields proportionally greater reduction. Second, TCS overhead remains negligible (2.57–8.79ms) regardless of model size. In contrast, ToMe's bipartite matching overhead is nearly fixed per layer, making it disproportionately expensive for small models (30.9% on DeiT-Tiny vs. 6.4% on DeiT-Base). Detailed block-wise latency profiling is provided in Appendix Table 25.

*Table 1.* Comparison with state-of-the-art methods on ImageNet-1K. Throughput measured on H100 GPU with batch size 128. Best results in **bold**. FFN TCS is applied *training-free* at inference time. FLOPs ↓ denotes percentage reduction from baseline. Speedup = compressed model throughput / baseline throughput.

| Model | Method | Top-1 (%) | Top-5 (%) | GFLOPs | FLOPs ↓ (%) | Throughput (img/s) | Speedup |
|---|---|---|---|---|---|---|---|
| DeiT-Tiny | Baseline | 72.20 | 91.10 | 1.3 | – | 2090.9 | 1.00× |
| | ToMe (Bolya et al., 2022) | 71.25 | 90.74 | **0.7** | **46.2** | 2484.6 | 1.19× |
| | DiffRate (Chen et al., 2023) | 71.78 | 90.87 | 0.9 | 30.8 | 2422.5 | 1.16× |
| | **ToaSt (Ours)** | **74.25** | **92.65** | 0.76 | 41.5 | **4249.7** | **2.03×** |
| DeiT-Small | Baseline | 79.82 | 94.95 | 4.6 | – | 2313.2 | 1.00× |
| | ToMe (Bolya et al., 2022) | 79.35 | 94.65 | 2.7 | 41.3 | 2737.1 | 1.18× |
| | DiffRate (Chen et al., 2023) | 79.56 | 94.80 | 2.9 | 37.0 | 2808.1 | 1.21× |
| | **ToaSt (Ours)** | **83.40** | **96.97** | **2.5** | **45.7** | **4783.3** | **2.07×** |
| DeiT-Base | Baseline | 81.80 | 95.60 | 17.6 | – | 1122.9 | 1.00× |
| | ToMe (Bolya et al., 2022) | 80.59 | 94.83 | 11.5 | 34.7 | 1628.4 | 1.45× |
| | DiffRate (Chen et al., 2023) | 81.51 | 95.40 | 11.5 | 34.7 | 1553.9 | 1.38× |
| | **ToaSt (Ours)** | **84.82** | **97.10** | **10.7** | **39.2** | **1690.9** | **1.51×** |
| ViT-MAE-Base | Baseline | 83.75 | **96.54** | 17.6 | – | 1140.2 | 1.00× |
| | ToMe (Bolya et al., 2022) (r=13) | 81.87 | 96.02 | **10.4** | **40.9** | **1783.3** | **1.56×** |
| | DiffRate (Chen et al., 2023) | 82.90 | 96.14 | 11.5 | 34.7 | 1552.8 | 1.36× |
| | **ToaSt (Ours)** | **84.13** | 96.39 | 11.0 | 37.5 | 1692.6 | 1.48× |
| ViT-MAE-Large | Baseline | 85.96 | 97.55 | 61.6 | – | 349.0 | 1.00× |
| | ToMe (Bolya et al., 2022) (r=6) | 84.58 | 97.12 | **38.5** | 37.5 | 523.2 | 1.50× |
| | DiffRate (Chen et al., 2023) | 85.66 | 97.44 | 42.3 | 31.3 | 474.7 | 1.36× |
| | **ToaSt (Ours)** | **88.94** | **97.95** | **38.5** | **37.5** | **527.0** | **1.51×** |
| ViT-MAE-Huge | Baseline | 86.88 | 98.07 | 167.4 | – | 129.7 | 1.00× |
| | ToMe (Bolya et al., 2022) (r=5) | 86.28 | 97.88 | 113.9 | 31.9 | 185.9 | 1.43× |
| | DiffRate (Chen et al., 2023) | 86.65 | 97.88 | 103.4 | 38.2 | 202.9 | 1.56× |
| | **ToaSt (Ours)** | **88.52** | **98.29** | **101.4** | **39.4** | **206.2** | **1.59×** |
| Swin-Tiny | Baseline | 81.20 | 95.50 | 4.5 | – | 2610.9 | 1.00× |
| | **ToaSt (Ours)** | **81.76** | **95.70** | **3.1** | **31.3** | **2705.8** | **1.04×** |
| Swin-Small | Baseline | 83.20 | 96.20 | 8.7 | – | 1534.4 | 1.00× |
| | STViT-R (Chang et al., 2023) | 82.60 | 96.07 | 5.8 | 33.3 | 1646.6 | 1.07× |
| | **ToaSt (Ours)** | **84.65** | **96.80** | **5.4** | **38.2** | **1909.5** | **1.24×** |
| Swin-Base | Baseline | 83.50 | **96.50** | 15.4 | – | 1100.1 | 1.00× |
| | STViT-R (Chang et al., 2023) | 83.20 | 96.40 | 10.3 | 33.1 | 1206.2 | 1.10× |
| | **ToaSt (Ours)** | **85.21** | **96.50** | **8.8** | **42.7** | **1408.6** | **1.28×** |

**Model Scale vs. Recovery Speed.** We observe a strong correlation between model scale and the efficiency of the fine-tuning phase. ViT-MAE-Huge requires only **15 epochs** to exceed baseline performance, whereas Large and Base versions require 139 and 297 epochs, respectively. This indicates that larger foundation models possess higher intrinsic redundancy, allowing for near-instantaneous recalibration after aggressive pruning.

### 4.3. Downstream Task Generalization

To validate that ToaSt removes genuine architectural redundancy rather than task-specific features, we evaluate on three downstream benchmarks: COCO object detection, ADE20K semantic segmentation, and CIFAR-100 classification.

Table 3 presents object detection results. Applying the same TCS configuration from ImageNet classification, Swin-Small achieves 52.2 box mAP compared to the baseline's

51.9 mAP—a +0.3 mAP improvement despite significant compression. Similarly, Swin-Base with 4-layer TCS improves to 52.2 mAP (+0.1 mask mAP), while more aggressive 6-layer compression maintains competitive performance at 51.8 box mAP and 44.9 mask mAP.

We further evaluate Swin-Base on ADE20K semantic segmentation (512 × 512) and CIFAR-100 classification (224 × 224) in Table 4. Applied training-free, TCS preserves ADE20K segmentation quality (0.00 mIoU drop with 4 layers, −0.15 with 6 layers), and CIFAR-100 accuracy slightly improves (+0.07%p) under full ToaSt, confirming robust cross-task generalization.

These results demonstrate that ToaSt's compression transfers effectively across diverse downstream tasks—object detection, semantic segmentation, and classification. The consistent maintenance or improvement of task performance despite significant FLOPs reduction supports our hypothesis

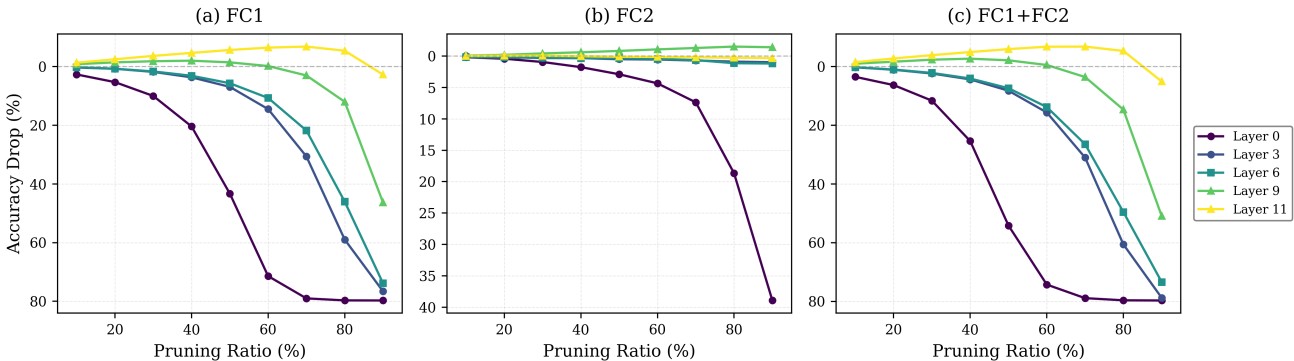

*Figure 5.* **Layer-wise FFN TCS Sensitivity Analysis.** Sensitivity analysis of FC1, FC2, and combined pruning across DeiT-Small layers at various ratios (10%-90%). (a) FC1 shows high sensitivity in early layers but robustness in later layers (L9-11), with L11 improving accuracy up to 80% pruning. (b) FC2 exhibits lower sensitivity, enabling aggressive pruning (50-90%) in later layers. (c) Combined pruning validates asymmetric layer-adaptive ratios exploiting distinct redundancy patterns between FC1 and FC2.

*Table 2.* Comparison at similar FLOPs budgets. ToaSt achieves +1–4%p higher accuracy than token compression methods.

| Model | Method | GFLOPs | Top-1 (%) | Δ Acc | Throughput(imgs/s) |
|---|---|---|---|---|---|
| DeiT Tiny | Baseline | 1.3 | 72.20 | – | 2090.95 |
| | DiffRate | 0.8 | 71.67 | - 0.53 | 2255.7 |
| | **ToaSt (Ours)** | 0.8 | **74.30** | **+2.1** | **4249.68** |
| DeiT Small | Baseline | 4.6 | 79.82 | – | 2313.2 |
| | ToMe | 2.7 | 79.35 | - 0.47 | 2737.05 |
| | DiffRate | 2.7 | 79.38 | - 0.44 | 3259.85 |
| | **ToaSt (Ours)** | 2.7 | **83.89** | **+4.07** | **4783.32** |
| DeiT Base | Baseline | 17.6 | 81.80 | – | 1122.89 |
| | DiffRate | 10.4 | 81.01 | - 0.79 | 1659.79 |
| | **ToaSt (Ours)** | 10.4 | **82.87** | **+1.07** | **1707.53** |
| ViT-MAE Large | Baseline | 61.6 | 85.96 | – | 349.03 |
| | ToMe (r=6) | 38.5 | 84.58 | - 1.38 | 523.2 |
| | DiffRate | 38.5 | 85.38 | - 0.58 | 513.26 |
| | **ToaSt (Ours)** | 38.5 | **88.94** | **+2.98** | **527.01** |
| ViT-MAE Huge | Baseline | 167.4 | 86.88 | – | 129.68 |
| | DiffRate | 103.4 | 86.65 | - 0.23 | 202.85 |
| | **ToaSt (Ours)** | 103.4 | **90.03** | **+3.15** | **206.21** |

that Token Channel Selection removes redundant features that act as noise, rather than discriminative features essential for downstream tasks.

### 4.4. Ablation Studies

**Coupled vs. Single-Sided Importance.** We compare our coupled importance formulation (computing scores from concatenated $[\mathbf{W}_Q; \mathbf{W}_K]$ and $[\mathbf{W}_V; \mathbf{W}_{\text{proj}}^\top]$) against single-sided alternatives that compute importance from one matrix and broadcast indices to the other. As shown in Appendix Table 15, the coupled formulation provides a consistent but modest improvement ($\leq 0.33$%p), confirming that the primary contribution is the index synchronization itself (Figure 3), with the concatenation offering an incremental benefit by considering both matrices jointly.

**Attention-Guided Weighting.** As detailed in §3.2, the attention-guided formulation (Eq. (6)) benefits CLS-distilled models (+2.2 to +8.0%p) but is neutral for others, validating our architecture-dependent design. Full results are in Appendix Table 16.

*Table 3.* Object detection results on COCO val2017 with Cascade Mask R-CNN. Our compressed backbones maintain or improve detection performance. Detailed configuration is in Appendix Table 26.

| Backbone | Method | GFLOPs | Box mAP | Mask mAP |
|---|---|---|---|---|
| Swin Small | Baseline | 194 | 51.9 | 45.0 |
| | **ToaSt** | 144 (25.8%↓) | **52.2** | 45.0 |
| Swin Base | Baseline | 343 | 51.9 | 45.0 |
| | **ToaSt** | 251 (26.8%↓) | **52.2** | 45.1 |
| | **ToaSt** | 246 (28.3%↓) | 51.8 | 44.9 |

*Table 4.* **Cross-task generalization on ADE20K and CIFAR-100** with Swin-Base. TCS applied training-free on ADE20K; CIFAR-100 uses full ToaSt with fine-tuning.

| Setting | GFLOPs | Quality | Δ |
|---|---|---|---|
| ADE20K (Baseline) | 87.0 | 48.13 mIoU | – |
| ADE20K (4-layer TCS) | 82.8 | 48.13 mIoU | 0.00 |
| ADE20K (6-layer TCS) | 80.9 | 47.98 mIoU | −0.15 |
| CIFAR-100 (Baseline) | 15.4 | 89.37% | – |
| CIFAR-100 (ToaSt) | 10.1 (34.4%↓) | 89.44% | +0.07 |

$\lambda$ **Sensitivity.** We evaluate the sensitivity of $\lambda_{cls}$ and $\lambda_{patch}$ in Eq. (6) across five CLS-token architectures (Appendix Table 17). Performance remains stable within $\lambda_{cls} \in [1, 3]$ ($\leq 1$%p variation), confirming that the default $\lambda_{cls} = 2.0, \lambda_{patch} = 1.0$ is a robust choice requiring no per-model tuning.

**Sampling Strategy.** We compare our default sampling rates against *No Sampling* (all $N$ tokens), aggressive $1\%$, and a single-token (*1 Sample*) setting across all nine architectures (Appendix Table 18). Default sampling achieves within $1.3$%p of full-token computation while delivering $\sim 3$–$5\%$ higher throughput; remarkably, even a single token preserves over $97\%$ of the full-token accuracy. This empirically validates the high channel-level linear dependency

*Table 5.* Ablation study on representative architectures. ToaSt consistently outperforms MHSA-only pruning across supervised (DeiT), self-supervised (ViT-MAE), and hierarchical (Swin) transformers. Full results in Appendix F.

| Model | Method | Top-1 (%) | GFLOPs | Throughput | Speedup |
|-------|--------|-----------|--------|------------|---------|
| DeiT Small | Baseline | 79.82 | 4.6 | 2313.20 | 1.00× |
| | MHSA Only | 76.12 | 3.1 | 3684.59 | 1.59× |
| | **ToaSt** | **83.40** | **2.5** | **4783.32** | **2.07×** |
| ViT-MAE Large | Baseline | 85.96 | 61.6 | 349.03 | 1.00× |
| | MHSA Only | 81.81 | 42.8 | 492.07 | 1.41× |
| | **ToaSt** | **88.94** | **38.5** | **527.01** | **1.51×** |
| Swin Base | Baseline | 83.50 | 15.4 | 1100.10 | 1.00× |
| | MHSA Only | 80.90 | 10.94 | 1345.46 | 1.22× |
| | **ToaSt** | **85.21** | **8.8** | **1408.60** | **1.28×** |

($R^2 > 0.9$, Section 3.2), confirming that channel importance can be reliably estimated from a tiny token subset.

**Component-wise Contribution.** Table 5 analyzes the individual contributions of MHSA structured pruning and training-free TCS across representative architectures. A consistent pattern emerges: MHSA-only pruning achieves moderate speedups (1.22–1.59×) but causes significant accuracy degradation (2.6–4.2%p). Adding TCS not only provides additional speedup but remarkably recovers and exceeds baseline accuracy by 1.7–3.6%p, demonstrating that the two components are complementary. This validates our decoupled design—MHSA pruning reduces attention computation while TCS removes redundant FFN channels that act as noise.

**Layer-wise Pruning Ratios.** Figure 5 examines the effect of different FC1/FC2 pruning configurations on DeiT-Small. We observe that asymmetric pruning (conservative FC1, aggressive FC2) consistently outperforms symmetric approaches, validating our analysis that FC1 performs information expansion while FC2 exhibits higher redundancy suitable for aggressive pruning.

**Pruning Ratio Pattern.** To verify that our structural principle—conservative FC1, aggressive FC2 in deeper layers (henceforth "fc2_heavy")—generalizes across architectures, we compare it against three alternatives: *uniform* (equal ratios on all layers), *stepwise* (gradually increasing with depth), and *inverse* (the reverse of ours). As shown in Appendix Table 19, fc2_heavy ranks 1st on 3/4 architectures and 2nd on the remaining one (MAE-Huge), confirming the principle is architecture-agnostic. Uniform and inverse patterns degrade severely (down to 18.06% Top-1 on DeiT-Small for inverse), corroborating our depth-dependent redundancy analysis (Section 3.2).

**Comparison with Low-Rank Methods.** Since our analysis identifies effective rank collapse in FFN layers (Section 3.2), we compare TCS against static low-rank decomposition (SVD) and learned adaptation (LoRA (Hu et al., 2022))

on DeiT-Small at matched FLOPs (Appendix Table 20). Truncated SVD (training-free) yields 69–73% Top-1, while rank-matched LoRA (20 epochs) reaches 76.13%—both substantially below TCS at 83.40% (training-free). A likely explanation is that our analysis measures rank collapse in the activation space (feature matrix $\mathbf{X}$, Section 3.2); TCS exploits this input-dependent structure by selecting channels from per-token activation magnitudes, whereas SVD and LoRA approximate weights with a fixed low-rank basis independent of inputs.

### 4.5. Compatibility with Token Compression

ToaSt operates on the channel dimension $D$, which is orthogonal to token compression methods that reduce sequence length $N$. To validate this complementarity, we combine ToaSt with ToMe (Bolya et al., 2022) across different model scales. The combination achieves additive throughput gains—e.g., DeiT-Small reaches 6138.1 img/s (2.65× baseline) at 1.83 GFLOPs, and ViT-MAE-Huge reaches 269.4 img/s at 74.5 GFLOPs—confirming that ToaSt's channel pruning is complementary to, not counteracted by, token compression. Full results are provided in Appendix Table 21.

## 5. Conclusion and Future Work

We presented ToaSt, a decoupled ViT compression framework combining structured coupled MHSA pruning with training-free Token Channel Selection for FFNs. Experiments across nine models and three downstream tasks (COCO, ADE20K, CIFAR-100) demonstrate strong generalization across DeiT, ViT-MAE, and Swin. Channel SNR analysis confirms TCS selectively removes noise-dominant channels, and ToaSt is orthogonal to token compression, yielding additive throughput gains. Current limitations include manual per-model tuning of layer-wise pruning ratios and their potential dataset dependence; while ImageNet-derived ratios transfer well to COCO and ADE20K, optimality is not guaranteed across all datasets. Future work includes learnable, dataset-adaptive ratio optimization, extension to vision-language models, and combination with quantization.

### Impact Statement

This paper presents work whose goal is to advance the field of machine learning. There are many potential societal consequences of our work, none of which we feel must be specifically highlighted here.

## Acknowledgements

This work was partly supported by the National Research Foundation of Korea(NRF) grant funded by the Korea government(MSIT) (RS-2026-25487368) and Hankuk University of Foreign Studies Research Fund (of 2025). This research was enabled in part by support provided by the Digital Research Alliance of Canada (https://alliancecan.ca). Cheonjun Park is the corresponding author.

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

# A. Background

## A.1. Vision Transformer Architecture

Vision Transformer (ViT) (Dosovitskiy, 2020) adapts the Transformer architecture (Vaswani et al., 2017) to computer vision by dividing an input image into non-overlapping patches and processing them as a sequence. Given an image $\mathbf{X} \in \mathbb{R}^{H \times W \times C}$ with patch size $P \times P$, ViT creates $N = \frac{HW}{P^2}$ patches, projects them into a $D$-dimensional embedding space, and adds positional embeddings.

The embedded sequence passes through $L$ stacked Transformer encoder blocks, each consisting of Multi-Head Self-Attention (MHSA) and Feed-Forward Network (FFN) with residual connections:

$$\mathbf{X}'_l = \text{MHSA}(\text{LN}(\mathbf{X}_{l-1})) + \mathbf{X}_{l-1} \tag{8}$$

$$\mathbf{X}_l = \text{FFN}(\text{LN}(\mathbf{X}'_l)) + \mathbf{X}'_l \tag{9}$$

**Multi-Head Self-Attention (MHSA)** computes attention across all tokens using $H$ parallel attention heads. For each head $h$, the input is projected into queries, keys, and values through learned weight matrices $\mathbf{W}_Q^h, \mathbf{W}_K^h, \mathbf{W}_V^h \in \mathbb{R}^{D \times d_k}$ where $d_k = D/H$. The attention mechanism computes pairwise similarities between all tokens, resulting in an $N \times N$ attention matrix that determines how information flows between tokens.

**Feed-Forward Network (FFN)** processes each token independently through two linear transformations with a GELU activation:

$$\text{FFN}(\mathbf{X}) = \text{GELU}(\mathbf{X}\mathbf{W}_{\text{FC1}})\mathbf{W}_{\text{FC2}} \tag{10}$$

where $\mathbf{W}_{\text{FC1}} \in \mathbb{R}^{D \times 4D}$ expands the hidden dimension by a factor of 4 (denoted as $D_{\text{mlp}} = 4D$), and $\mathbf{W}_{\text{FC2}} \in \mathbb{R}^{4D \times D}$ projects back to the original dimension. This expansion-contraction pattern is standard across Transformer architectures.

## A.2. Computational Complexity and Model Scaling

The computational cost of Vision Transformers stems from two distinct sources: operations that scale with sequence length $N$ (primarily attention), and operations that scale with hidden dimensions $D$ (projections and FFN). Table 6 summarizes the complexity and FLOPs distribution of each operation.

*Table 6.* Complexity breakdown of ViT component operations (DeiT-Small: $N$=197, $D$=384).

| Operation | Complexity | FLOPs (%) | Primary Factor |
|---|---|---|---|
| $Q, K, V$ Projections | $\mathcal{O}(3ND^2)$ | ~23.0% | Hidden dim $D$ |
| Attention Score ($QK^\top$) | $\mathcal{O}(N^2D)$ | ~3.9% | Sequence length $N$ |
| Attention Output ($AV$) | $\mathcal{O}(N^2D)$ | ~3.9% | Sequence length $N$ |
| Output Projection | $\mathcal{O}(ND^2)$ | ~7.7% | Hidden dim $D$ |
| FC1 | $\mathcal{O}(4ND^2)$ | ~30.7% | Hidden dim $D$ |
| FC2 | $\mathcal{O}(4ND^2)$ | ~30.7% | Hidden dim $D$ |

The MHSA block contributes approximately 38.5% of total FLOPs, with the $\mathcal{O}(N^2D)$ pairwise similarity computation ($QK^\top$ and $AV$) accounting for ~7.8% and the $Q, K, V$ and output projections contributing the remaining ~30.7%. However, FFN layers dominate the computational cost, accounting for roughly 61% of FLOPs due to the large matrix multiplications with $\mathbf{W}_{\text{FC1}}$ and $\mathbf{W}_{\text{FC2}}$, each requiring $4ND^2$ operations.

**Scaling Trends.** Recent scaling studies (Zhai et al., 2022; Dehghani et al., 2023) have identified width ($D$) and MLP dimension ($D_{\text{mlp}}$) as primary drivers of model capacity. Figure 6 illustrates the scaling trends of standard ViT configurations, ranging from ViT-Ti (5.7M parameters) to the massive ViT-22B.

As clearly shown in the right panel of Figure 6, while the number of layers tends to plateau after ViT-L, the model width ($D$) increases substantially to drive parameter growth. This specific scaling trend has direct implications for computational cost distribution. As models scale wider, the FFN's relative contribution increases—from 61% in ViT-B to 72% in ViT-22B. This shift occurs because FFN computation scales with $D \times D_{\text{mlp}}$ (where $D_{\text{mlp}} = 4D$), while attention scales with $N^2D$. Consequently, as width becomes the dominant scaling factor, the FFN becomes increasingly dominant in the computational profile.

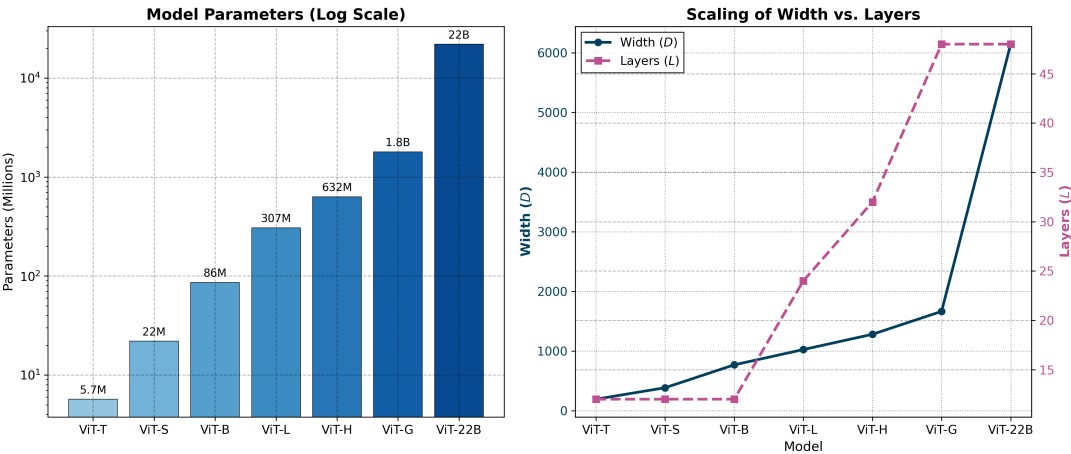

*Figure 6.* Scaling trends of standard ViT models. (Left) Total parameter count. (Right) Model width ($D$) vs. depth ($L$). Width increases dramatically while depth saturates in larger models.

These two sources of complexity—sequence length and hidden dimensions—motivate complementary compression strategies: token pruning to reduce $N$ and address attention costs, and channel/weight pruning to reduce $D$ and address FFN costs. Our work focuses on the latter through structured pruning of both MHSA weights and FFN channels.

## B. ToaSt Pipeline Pseudocode

---

**Algorithm 1** ToaSt Inference Pipeline

---

**Require:** Pretrained ViT with $L$ layers, MHSA ratio $\rho_m$, FFN ratios $\{r_{\text{FC1}}^{(l)}, r_{\text{FC2}}^{(l)}\}$; input $\mathbf{X} \in \mathbb{R}^{N \times D}$; $\mathbf{W}_{\text{FC1}} \in \mathbb{R}^{D \times 4D}$, $\mathbf{W}_{\text{FC2}} \in \mathbb{R}^{4D \times D}$
1: **// Offline: Structured Coupled Weight Pruning (SCWP)**
2: **for** each layer $l = 1, \ldots, L$ **do**
3:     **for** each head $h = 1, \ldots, H$ **do**
4:         $\mathbf{W}_{QK}^h = [\mathbf{W}_Q^h; \mathbf{W}_K^h]$, $\mathbf{W}_{VO}^h = [\mathbf{W}_V^h; (\mathbf{W}_{\text{proj}}^h)^\top]$
5:         $I_{QK}^h[j] = \|\mathbf{w}_{QK,j}^h - \text{GM}(\mathbf{W}_{QK}^h)\|_2$, $I_{VO}^h[j] = \|\mathbf{w}_{VO,j}^h - \text{GM}(\mathbf{W}_{VO}^h)\|_2$
6:         $\mathcal{K}_l^h = \text{TopK}(\frac{1}{2}(I_{QK}^h + I_{VO}^h), d_k')$, where $d_k' = \lfloor (1 - \rho_m)d_k \rfloor$ (uniform across heads)
7:     **end for**
8:     Gather columns of $\mathbf{W}_Q^h, \mathbf{W}_K^h, \mathbf{W}_V^h$ and rows of $\mathbf{W}_{\text{proj}}^h$ at $\mathcal{K}_l^h$ ($d_k \to d_k'$)
9: **end for**
10: **// Online: Forward Pass with Token Channel Selection (TCS); token count $N$ preserved**
11: **for** each layer $l = 1, \ldots, L$ **do**
12:     $\mathbf{X}' \leftarrow \text{MHSA}_{\text{pruned}}(\text{LN}(\mathbf{X})) + \mathbf{X}$; $\mathbf{Z} \leftarrow \text{LN}(\mathbf{X}') \in \mathbb{R}^{N \times D}$
13:     Sample token subset $\mathcal{S} \subset \{1, \ldots, N\}, |\mathcal{S}| \in [0.02N, 0.2N]$
14:     **// FC1: channel selection on input dim $D$ (channel axis of Z)**
15:     Compute $\mathbf{I}_{\text{FC1}} \in \mathbb{R}^D$ via Eq. (6) or Eq. (7) over $\mathcal{S}$
16:     $\mathcal{C}_1^{(l)} = \text{TopK}(\mathbf{I}_{\text{FC1}}, k_1), k_1 = \lfloor (1 - r_{\text{FC1}}^{(l)})D \rfloor$
17:     Gather columns of $\mathbf{Z}$ and rows of $\mathbf{W}_{\text{FC1}}$ at $\mathcal{C}_1^{(l)}$: $\mathbf{Z}_{\mathcal{C}_1} \in \mathbb{R}^{N \times k_1}$, $\mathbf{W}_{\text{FC1}}[\mathcal{C}_1^{(l)}, :] \in \mathbb{R}^{k_1 \times 4D}$
18:     $\mathbf{H} \leftarrow \text{GELU}(\mathbf{Z}_{\mathcal{C}_1} \mathbf{W}_{\text{FC1}}[\mathcal{C}_1^{(l)}, :]) \in \mathbb{R}^{N \times 4D}$
19:     **// FC2: channel selection on expanded dim $4D$ (channel axis of H)**
20:     Compute $\mathbf{I}_{\text{FC2}} \in \mathbb{R}^{4D}$ over $\mathbf{H}$ using the same $\mathcal{S}$ (Eq. (6) or Eq. (7))
21:     $\mathcal{C}_2^{(l)} = \text{TopK}(\mathbf{I}_{\text{FC2}}, k_2), k_2 = \lfloor (1 - r_{\text{FC2}}^{(l)}) \cdot 4D \rfloor$
22:     Gather columns of $\mathbf{H}$ and rows of $\mathbf{W}_{\text{FC2}}$ at $\mathcal{C}_2^{(l)}$: $\mathbf{H}_{\mathcal{C}_2} \in \mathbb{R}^{N \times k_2}$, $\mathbf{W}_{\text{FC2}}[\mathcal{C}_2^{(l)}, :] \in \mathbb{R}^{k_2 \times D}$
23:     $\mathbf{X} \leftarrow \mathbf{H}_{\mathcal{C}_2} \mathbf{W}_{\text{FC2}}[\mathcal{C}_2^{(l)}, :] + \mathbf{X}'$
24: **end for**
25: **return** $\mathbf{X} \in \mathbb{R}^{N \times D}$

---

## C. Importance Metric Comparison

Table 7 compares importance metrics for structured MHSA pruning across architectures (first layer skipped).

*Table 7.* Comparison of importance metrics across architectures.

| Model | $L_1$-norm | $L_2$-norm | Geometric Median |
|---|---|---|---|
| DeiT-Small (90%) | 76.12 | 76.17 | **76.24** |
| Swin-Small (90%) | 80.54 | 81.14 | **81.20** |

The performance gap across metrics is marginal ($\leq 0.66\%$), indicating that MHSA structured pruning is robust to the choice of importance metric. The geometric median achieves the highest accuracy by identifying functionally redundant dimensions whose representations can be recovered from others, validating its adoption following FPGM (He et al., 2019).

## D. Channel SNR Analysis

To explain the accuracy improvement after FFN pruning, we analyze the signal-to-noise ratio (SNR) of kept vs. pruned channels in FC2-pruned blocks of Swin-Base. Table 8 shows that pruned channels consistently exhibit 3–5.5× lower SNR, confirming that TCS selectively removes noise-dominant channels.

*Table 8.* **Channel SNR analysis on Swin-Base.** Kept vs. pruned channel SNR in FC2-pruned blocks.

| Block | Kept SNR | Pruned SNR | Ratio |
|---|---|---|---|
| 17 | 0.198 | 0.065 | 3.04× |
| 18 | 0.229 | 0.055 | 4.20× |
| 19 | 0.282 | 0.051 | 5.50× |
| 20 | 0.409 | 0.077 | 5.29× |
| 21 | 0.562 | 0.153 | 3.67× |

## E. Detailed Experimental Configurations

This appendix provides the layer-wise pruning configurations used in our experiments. We present representative configurations for one model per family (DeiT-Small, ViT-MAE-Huge, Swin-Base). For all models, we apply structured coupled weight pruning to MHSA and adaptive training-free Token Channel Selection (TCS) to FFN layers. The FFN pruning ratios are determined adaptively at inference time based on activation patterns, requiring no additional training or calibration. Full per-layer configurations for all nine models are available in our codebase.

*Table 9.* **Computational reduction in MHSA with 90% per-head pruning.** The reduction applies uniformly across all linear and attention operations.

| Operation | Original FLOPs | Pruned FLOPs | Reduction |
|---|---|---|---|
| $Q, K$ Projection | $2ND \cdot d_k H$ | $2ND \cdot 0.1 d_k H$ | **90%** |
| $QK^\top$ (Attn) | $N^2 d_k H$ | $N^2 \cdot 0.1 d_k H$ | **90%** |
| $V$ Projection | $ND \cdot d_v H$ | $ND \cdot 0.1 d_v H$ | **90%** |
| $AV$ (Weighted Sum) | $N^2 d_v H$ | $N^2 \cdot 0.1 d_v H$ | **90%** |
| Output Projection | $N d_v H \cdot D$ | $N \cdot 0.1 d_v H \cdot D$ | **90%** |

*Table 10.* **Fine-tuning cost for MHSA pruning.** Measured on 4× H100 GPUs. TCS requires no training.

| Model | Epochs | Wall-clock Time |
|---|---|---|
| DeiT-Small | 290 | ~1 day |
| ViT-MAE-Large | 139 | ~31 hours |
| ViT-MAE-Huge | 15 | ~15 hours |

*Table 11.* Layer-wise FFN pruning ratios for DeiT-Small (12 layers). MHSA: 90% channel reduction (except layer 1).

| Layer | 1 | 2 | 3 | 4 | 5 | 6 | 7 | 8 | 9 | 10 | 11 | 12 |
|---|---|---|---|---|---|---|---|---|---|---|---|---|
| FC1 | 0 | 0 | 0 | 0 | 0 | 0 | 0 | 0 | 0 | 0 | 0.5 | 0.5 |
| FC2 | 0 | 0 | 0 | 0 | 0 | 0 | 0 | 0.8 | 0.8 | 0.9 | 0.9 | 0.9 |

*Table 12.* Layer-wise FFN pruning ratios for ViT-MAE-Huge (32 layers). MHSA: 90% channel reduction (except layer 1). Only FC2 is pruned in the final 8 layers.

| Layer | 1-24 | 25 | 26 | 27 | 28 | 29 | 30 | 31 | 32 |
|---|---|---|---|---|---|---|---|---|---|
| FC1 | 0 | 0 | 0 | 0 | 0 | 0 | 0 | 0 | 0.2 |
| FC2 | 0 | 0.8 | 0.9 | 0.9 | 0.9 | 0.9 | 0.9 | 0.9 | 0.9 |

*Table 13.* Layer-wise FFN pruning ratios for Swin-Base. Stage configuration: [2, 2, 18, 2] blocks. MHSA: 90% channel reduction (except first layer of each stage).

| | Stage 0 | | Stage 1 | | Stage 2 | | | | | | | | | | | | | | | | | | Stage 3 | |
|---|---|---|---|---|---|---|---|---|---|---|---|---|---|---|---|---|---|---|---|---|---|---|---|---|
| Layer | 1 | 2 | 1 | 2 | 1 | 2 | 3 | 4 | 5 | 6 | 7 | 8 | 9 | 10 | 11 | 12 | 13 | 14 | 15 | 16 | 17 | 18 | 1 | 2 |
| FC1 | 0 | 0 | 0 | 0 | 0 | 0 | 0 | 0 | 0 | 0 | 0 | 0 | 0 | 0 | 0 | 0 | 0 | 0 | 0 | 0 | 0 | 0 | 0 | 0.3 |
| FC2 | 0 | 0 | 0 | 0 | 0 | 0 | 0 | 0 | 0 | 0 | 0 | 0 | 0 | 0.9 | 0.9 | 0.9 | 0.9 | 0.9 | 0.9 | 0.9 | 0.9 | 0.9 | 0.9 | 0.9 |

# F. Full Ablation Results

This section consolidates all ablation studies supporting the design choices in our method. We organize results into seven subsections: (1) component-wise contribution across all architectures, (2) coupled vs. single-sided importance for SCWP, (3) attention-guided weighting for FFN importance, (4) $\lambda$ sensitivity analysis, (5) sampling strategy ablation, (6) pruning ratio pattern ablation, and (7) comparison with low-rank methods.

### F.1. Component-wise Contribution (Full)

Table 14 extends the representative results in Table 5 to all nine architectures. ToaSt consistently outperforms MHSA-only pruning, confirming the complementarity of the two components.

*Table 14.* Complete ablation study across all evaluated architectures.

| Model | Method | Top-1 (%) | GFLOPs | Throughput | Speedup |
|---|---|---|---|---|---|
| | Baseline | 72.20 | 1.3 | 2090.95 | 1.00× |
| DeiT-Tiny | MHSA Only | 69.81 | 0.9 | 2947.26 | 1.41× |
| | **ToaSt** | **74.25** | **0.76** | **4249.68** | **2.03×** |
| | Baseline | 79.82 | 4.6 | 2313.20 | 1.00× |
| DeiT-Small | MHSA Only | 76.12 | 3.1 | 3684.59 | 1.59× |
| | **ToaSt** | **83.40** | **2.5** | **4783.32** | **2.07×** |
| | Baseline | 81.80 | 17.6 | 1122.89 | 1.00× |
| DeiT-Base | MHSA Only | 78.08 | 12.4 | 1569.49 | 1.40× |
| | **ToaSt** | **82.87** | **10.4** | **1707.53** | **1.52×** |
| | Baseline | 83.75 | 17.6 | 1140.16 | 1.00× |
| ViT-MAE-Base | MHSA Only | 79.74 | 12.4 | 1569.10 | 1.38× |
| | **ToaSt** | **84.13** | **11.0** | **1692.61** | **1.48×** |
| | Baseline | 85.96 | 61.6 | 349.03 | 1.00× |
| ViT-MAE-Large | MHSA Only | 81.81 | 42.8 | 492.07 | 1.41× |
| | **ToaSt** | **88.94** | **38.5** | **527.01** | **1.51×** |
| | Baseline | 86.88 | 167.4 | 129.68 | 1.00× |
| ViT-MAE-Huge | MHSA Only | 82.52 | 115.6 | 187.98 | 1.45× |
| | **ToaSt** | **88.52** | **101.4** | **206.21** | **1.59×** |
| | Baseline | 81.20 | 4.5 | 2610.94 | 1.00× |
| Swin-Tiny | MHSA Only | 79.59 | 3.4 | 2529.68 | 0.97× |
| | **ToaSt** | **81.76** | **3.1** | **2705.76** | **1.04×** |
| | Baseline | 83.20 | 8.7 | 1534.41 | 1.00× |
| Swin-Small | MHSA Only | 80.54 | 6.2 | 1804.92 | 1.18× |
| | **ToaSt** | **84.65** | **5.4** | **1909.46** | **1.24×** |
| | Baseline | 83.50 | 15.4 | 1100.10 | 1.00× |
| Swin-Base | MHSA Only | 80.90 | 10.94 | 1345.46 | 1.22× |
| | **ToaSt** | **85.21** | **8.8** | **1408.60** | **1.28×** |

## F.2. Coupled vs. Single-Sided Importance

*Table 15.* **Ablation on importance source for SCWP.** DeiT-Small with 90% head-dimension pruning, 50 epochs fine-tuning.

| Importance Source | Top-1 (%) | Top-5 (%) |
|---|---|---|
| **Coupled (Ours)** | **71.43** | **90.75** |
| Q only → broadcast to K | 71.28 | 90.53 |
| K only → broadcast to Q | 71.32 | 90.71 |
| Proj only → broadcast to V | 71.10 | 90.38 |

## F.3. Attention-Guided Weighting

*Table 16.* **Effect of attention-guided weighting** ($\lambda_{cls}$ in Eq. (6)) on Top-1 accuracy across all architectures.

| Model | | With Attn (%) | Without Attn (%) | $\Delta$ |
|---|---|---|---|---|
| | Tiny | 74.25 | 66.22 | +8.03 |
| DeiT | Small | 83.40 | 75.83 | +7.57 |
| | Base | 84.82 | 82.60 | +2.22 |
| | Large | 89.21 | 89.51 | −0.30 |
| ViT-MAE | Huge | 88.52 | 90.52 | −2.00 |
| | Tiny | 81.76 | 81.85 | −0.09 |
| Swin | Small | 85.21 | 84.61 | +0.60 |
| | Base | 85.21 | 85.21 | 0.00 |

## F.4. $\lambda$ Sensitivity Analysis

*Table 17.* **Sensitivity of** $\lambda_{cls}$ **and** $\lambda_{patch}$ in Eq. (6). Top-1 accuracy across CLS-token architectures. Best per model in **bold**.

| $\lambda_{cls}$ | $\lambda_{patch}$ | DeiT-T | DeiT-S | DeiT-B | MAE-B | MAE-H |
|---|---|---|---|---|---|---|
| 0.0 | 1.0 | 65.54 | 75.39 | 83.26 | **85.40** | **90.57** |
| 1.0 | 1.0 | **74.58** | 83.37 | 84.75 | 84.94 | 89.50 |
| **2.0** | **1.0** | 74.25 | **83.40** | 84.82 | 84.13 | 88.52 |
| 3.0 | 1.0 | 73.51 | 83.13 | **85.03** | 83.01 | 87.33 |
| 5.0 | 1.0 | 72.31 | 82.84 | 84.84 | 80.98 | 85.12 |
| 1.0 | 2.0 | 73.85 | 82.40 | 84.40 | 85.45 | 89.90 |

## F.5. Sampling Strategy Ablation

To validate the design of our token sampling strategy (Section 3.2, Step.1), we compare four configurations across all nine architectures: (i) *No Sampling* (all $N$ tokens), (ii) our default layer-adaptive rate (2% for DeiT-S/B and the ViT-MAE family; 20% for DeiT-Tiny and the Swin variants, which operate on fewer tokens per stage), (iii) an aggressive 1% rate, and (iv) the minimal single-token (*1 Sample*) setting.

Three observations emerge from Table 18: (1) The default sampling rate matches *No Sampling* within 1.3%p Top-1 while delivering 3–5% higher throughput, since reduced selection overhead outweighs the marginal accuracy cost. (2) Aggressive rates (1% and *1 Sample*) preserve over 97% of full-token accuracy on most models, empirically validating the high channel-level linear dependency ($R^2 > 0.9$, Section 3.2). (3) Throughput is nearly saturated at the default rate, indicating that further reduction yields negligible efficiency gains while incurring measurable accuracy degradation. These results justify our choice of default sampling rates as a Pareto-optimal operating point.

*Table 18.* **Sampling strategy ablation across nine ViT architectures.** Top-1 accuracy (%) and throughput (img/s) on ImageNet-1K (H100 GPU, batch size 128). Default configurations used in the main paper are in **bold**.

| Model | Sampling | Top-1 | GFLOPs | Throughput | Model | Sampling | Top-1 | GFLOPs | Throughput |
|---|---|---|---|---|---|---|---|---|---|
| DeiT-Tiny | No Sampling | 75.21 | 0.76 | 4100.7 | ViT-MAE-Huge | No Sampling | 88.64 | 101.4 | 201.3 |
| | **20% (default)** | **74.25** | **0.76** | **4249.7** | | **2% (default)** | **88.52** | **101.4** | **206.2** |
| | 1% | 72.65 | 0.76 | 4228.5 | | 1% | 88.06 | 101.4 | 206.0 |
| | 1 Sample | 72.65 | 0.76 | 4251.5 | | 1 Sample | 87.71 | 101.4 | 206.3 |
| DeiT-Small | No Sampling | 84.67 | 2.5 | 4564.2 | Swin-Tiny | No Sampling | 82.12 | 2.5 | 3073.6 |
| | **2% (default)** | **83.40** | **2.5** | **4783.3** | | **20% (default)** | **81.76** | **2.5** | **3096.2** |
| | 1% | 80.77 | 2.5 | 4982.1 | | 2% | 77.97 | 2.5 | 3104.7 |
| | 1 Sample | 80.77 | 2.5 | 4871.7 | | 1 Sample | 77.85 | 2.5 | 3107.5 |
| DeiT-Base | No Sampling | 85.31 | 10.7 | 1631.7 | Swin-Small | No Sampling | 84.62 | 5.4 | 1871.3 |
| | **2% (default)** | **84.82** | **10.7** | **1690.9** | | **20% (default)** | **84.65** | **5.4** | **1909.5** |
| | 1% | 84.19 | 10.7 | 1691.2 | | 2% | 82.58 | 5.4 | 1916.6 |
| | 1 Sample | 84.19 | 10.7 | 1690.2 | | 1 Sample | 81.65 | 5.4 | 1916.1 |
| ViT-MAE-Base | No Sampling | 85.36 | 11.0 | 1618.1 | Swin-Base | No Sampling | 85.26 | 8.8 | 1370.4 |
| | **2% (default)** | **84.13** | **11.0** | **1692.6** | | **20% (default)** | **85.21** | **8.8** | **1408.6** |
| | 1% | 81.66 | 11.0 | 1693.3 | | 2% | 84.27 | 8.8 | 1426.0 |
| | 1 Sample | 81.66 | 11.0 | 1695.4 | | 1 Sample | 82.76 | 8.8 | 1428.4 |
| ViT-MAE-Large | No Sampling | 89.21 | 38.5 | 511.6 | | | | | |
| | **2% (default)** | **88.94** | **38.5** | **527.0** | | | | | |
| | 1% | 88.36 | 38.5 | 528.3 | | | | | |
| | 1 Sample | 88.36 | 38.5 | 528.4 | | | | | |

## F.6. Pruning Ratio Pattern Ablation

We validate that our pruning ratio pattern—*fc2_heavy* (conservative FC1, aggressive FC2 in deeper layers)—generalizes across architectures by comparing against three alternative depth-dependent patterns under matched FLOPs budgets:

- **fc2_heavy (Ours)**: FC1 conservative throughout; FC2 aggressively pruned in deeper layers, following our depth-

dependent redundancy analysis (Section 3.2).

- **uniform**: identical pruning ratio applied uniformly to all FFN layers.

- **stepwise**: ratios increase gradually (stepwise) with layer depth, applied symmetrically to FC1 and FC2.

- **inverse**: the reverse of fc2_heavy—FC1 aggressively pruned in deeper layers, FC2 conservative.

Table 19 reports Top-1 accuracy across four representative architectures. Our *fc2_heavy* pattern ranks 1st on DeiT-Small, ViT-MAE-Large, and Swin-Base, and 2nd on ViT-MAE-Huge (where *stepwise* achieves slightly higher accuracy, 90.06% vs. 88.79%). The *uniform* and *inverse* patterns degrade catastrophically (down to 18.06% on DeiT-Small for *inverse*), confirming that depth-aware asymmetric allocation is essential. These results validate our analysis-driven principle as a robust, architecture-agnostic default, while acknowledging that fine-grained per-layer ratios may still benefit from model-specific sensitivity sweeps ($\sim$1 hour per model on a single GPU).

*Table 19.* **Pruning ratio pattern ablation.** Top-1 accuracy (%) on ImageNet-1K across four representative architectures under matched FLOPs budgets. Best per model in **bold**, second-best underlined.

| Pattern | DeiT-S | MAE-L | MAE-H | Swin-B |
|---|---|---|---|---|
| **fc2_heavy (Ours)** | **82.84** | **88.24** | 88.79 | **85.85** |
| stepwise | 81.01 | 86.16 | **90.06** | 84.29 |
| uniform | 56.54 | 77.56 | 77.04 | 74.42 |
| inverse | 18.06 | 74.57 | 68.24 | 63.06 |

### F.7. Comparison with Low-Rank Methods

Our FFN redundancy analysis (Section 3.2) identifies effective rank collapse as a key motivation for channel selection. A natural alternative would be to exploit this collapse through low-rank weight decomposition—either statically via Singular Value Decomposition (SVD) or through learned adaptations such as LoRA (Hu et al., 2022). We compare TCS against both baselines on DeiT-Small at matched FLOPs in Table 20.

Both low-rank baselines substantially underperform TCS. Truncated SVD applied directly to FC1/FC2 weights yields 69–73% Top-1 (training-free), while rank-matched LoRA fine-tuned for 20 epochs reaches only 76.13%—a 7.3%p gap below TCS at 83.40% (training-free).

The key distinction lies in *where* the redundancy resides. SVD and LoRA assume that redundancy is a static property of weight matrices and can be captured by a fixed low-rank approximation. In contrast, our analysis (Section 3.2) shows that the rank collapse is an *activation-level* phenomenon: which channels are informative varies across tokens and across input images. TCS dynamically adapts to these per-token activation patterns at inference time, exploiting redundancy that static weight decomposition cannot reach. This explains the substantial performance gap and confirms that FFN compression benefits from input-adaptive selection rather than fixed weight-space approximation.

*Table 20.* **Comparison with low-rank methods on DeiT-Small at matched FLOPs.** TCS substantially outperforms both training-free SVD and learned LoRA, validating that FFN redundancy is best exploited through token-adaptive channel selection rather than fixed or learned weight decomposition.

| Method | Training | Top-1 (%) | $\Delta$ vs. TCS |
|---|---|---|---|
| SVD (truncated) | Free | 69–73 | $-10$ to $-15$ |
| LoRA (rank-matched, 20 ep.) | 20 epochs | 76.13 | $-7.27$ |
| **TCS (Ours)** | **Free** | **83.40** | — |

## G. Compatibility with Token Compression

ToaSt operates on the channel dimension $D$ while token compression methods reduce sequence length $N$; the two are orthogonal and can be directly combined to yield additive throughput gains, as shown in Table 21.

*Table 21.* Combination of ToaSt with ToMe token compression. ToaSt and ToMe target orthogonal dimensions (channel $D$ vs. sequence $N$), enabling direct combination with additive throughput gains.

| Model | Method | Top-1 (%) | Top-5 (%) | GFLOPs | Throughput (img/s) |
|---|---|---|---|---|---|
| DeiT-Tiny | ToMe | 71.25 | 90.74 | 0.7G | 2484.6 |
| | ToaSt | 74.25 | 92.65 | 0.76G | 4249.7 |
| | ToaSt + ToMe (r=7) | 72.30 | – | 0.64G | 3636.0 |
| DeiT-Small | ToMe | 79.35 | 94.65 | 2.7G | 2737.1 |
| | ToaSt | 83.40 | 96.97 | 2.5G | 4783.3 |
| | ToaSt + ToMe (r=13) | 79.98 | 95.64 | 1.83G | 6138.1 |
| DeiT-Base | ToMe | 80.59 | 94.83 | 11.5G | 1628.4 |
| | ToaSt | 84.82 | 97.10 | 10.7G | 1690.9 |
| | ToaSt + ToMe (r=11) | 82.36 | 96.10 | 8.0G | 2614.0 |
| ViT-MAE-Base | ToMe (r=13) | 81.87 | 96.02 | 10.4G | 1783.3 |
| | ToaSt | 84.13 | 96.39 | 11.0G | 1692.6 |
| | ToaSt + ToMe (r=7) | 83.57 | 96.04 | 9.17G | 1862.4 |
| ViT-MAE-Large | ToMe (r=6) | 84.58 | 97.12 | 38.5G | 523.2 |
| | ToaSt | 88.94 | 97.95 | 38.5G | 527.0 |
| | ToaSt + ToMe (r=6) | 87.90 | 97.61 | 26.54G | 711.0 |
| ViT-MAE-Huge | ToMe | 84.58 | – | 113.9G | 185.9 |
| | ToaSt | 88.52 | 98.29 | 101.4G | 206.2 |
| | ToaSt + ToMe (r=5) | 87.54 | 97.93 | 74.5G | 269.4 |

## H. Detailed Latency and Throughput Results

Tables 22–24 present comprehensive latency and throughput measurements for all models and configurations on H100 GPU with batch size 128.

*Table 22.* Detailed results for DeiT models on ImageNet-1K (H100 GPU, batch size 128).

| Model | Method | Top-1 (%) | Top-5 (%) | GFLOPs | FLOPs ↓ (%) | Latency (ms) | Throughput (img/s) | Speedup |
|---|---|---|---|---|---|---|---|---|
| DeiT-Tiny | Baseline | 72.20 | 91.10 | 1.3 | – | 61.22 | 2090.95 | – |
| | ToMe | 71.25 | 90.74 | 0.7 | 46.2 | 51.54 | 2484.59 | 1.19× |
| | DiffRate | 71.78 | 90.87 | 0.9 | 30.8 | 52.84 | 2422.54 | 1.16× |
| | DiffRate | 71.67 | 90.78 | 0.8 | 38.5 | 56.75 | 2255.70 | 1.08× |
| | ToaSt | 69.93 | 89.67 | 0.9 | 30.8 | 36.89 | 3469.48 | 1.66× |
| | ToaSt | 74.30 | 92.26 | 0.8 | 38.5 | 29.32 | 4364.95 | 2.09× |
| | ToaSt | **74.25** | **92.65** | 0.76 | 41.5 | 30.12 | 4249.68 | 2.03× |
| DeiT-Small | Baseline | 79.82 | 94.95 | 4.6 | – | 55.33 | 2313.20 | – |
| | ToMe | 79.35 | 94.65 | 2.7 | 41.3 | 44.24 | 2737.05 | 1.18× |
| | DiffRate | 79.56 | 94.80 | 2.9 | 37.0 | 45.58 | 2808.05 | 1.21× |
| | DiffRate | 79.38 | 94.65 | 2.7 | 41.3 | 39.27 | 3259.82 | 1.41× |
| | DiffRate | 79.09 | 94.50 | 2.5 | 45.7 | 39.30 | 3256.77 | 1.41× |
| | ToaSt | 80.77 | 95.29 | 2.9 | 37.0 | 28.32 | 4519.28 | 1.95× |
| | ToaSt | 83.89 | 97.13 | 2.7 | 41.3 | 27.47 | 4659.27 | 2.01× |
| | ToaSt | **83.40** | 96.97 | 2.5 | 45.7 | 26.76 | **4783.32** | **2.07×** |
| DeiT-Base | Baseline | 81.80 | 95.60 | 17.6 | – | 113.99 | 1122.89 | – |
| | ToMe | 80.59 | 94.83 | 11.5 | 34.7 | 78.60 | 1628.40 | 1.45× |
| | DiffRate | 81.01 | 95.02 | 10.4 | 40.9 | 75.48 | 1659.79 | 1.48× |
| | DiffRate | 81.51 | 95.40 | 11.5 | 34.7 | 82.37 | 1553.88 | 1.38× |
| | ToaSt | 82.25 | 96.07 | 11.5 | 34.7 | 78.54 | 1620.86 | 1.44× |
| | ToaSt | **84.82** | **97.10** | 10.7 | 39.2 | 75.70 | 1690.93 | 1.51× |
| | ToaSt | 82.87 | 96.29 | 10.4 | 40.9 | 74.96 | 1707.53 | 1.52× |
| | ToaSt | 82.02 | 95.57 | 10.27 | 41.6 | 74.83 | 1710.59 | 1.52× |

*Table 23.* Detailed results for ViT-MAE models on ImageNet-1K (H100 GPU, batch size 128).

| Model | Method | Top-1 (%) | Top-5 (%) | GFLOPS | FLOPs ↓ (%) | Latency (ms) | Throughput (img/s) | Speedup |
|---|---|---|---|---|---|---|---|---|
| ViT-MAE-Base | Baseline | 83.75 | 96.54 | 17.6 | – | 112.27 | 1140.16 | – |
| | ToMe (r=16) | 81.09 | 95.61 | 8.8 | 50.0 | 62.00 | 2064.49 | 1.81× |
| | ToMe (r=13) | 81.87 | 96.02 | 10.4 | 40.9 | 71.78 | 1783.33 | 1.56× |
| | DiffRate | 82.90 | 96.14 | 11.5 | 34.7 | 82.43 | 1552.75 | 1.36× |
| | ToaSt | 83.44 | 95.49 | 11.5 | 34.7 | 78.80 | 1624.28 | 1.42× |
| | ToaSt | **84.13** | 96.39 | 11.0 | 37.5 | 75.62 | 1692.61 | 1.48× |
| ViT-MAE-Large | Baseline | 85.96 | 97.55 | 61.6 | – | 366.73 | 349.03 | – |
| | ToMe (r=8) | 83.44 | 97.00 | 31.0 | 49.7 | 200.37 | 638.82 | 1.83× |
| | ToMe (r=6) | 84.58 | 97.12 | 38.5 | 37.5 | 244.65 | 523.20 | 1.50× |
| | DiffRate | 85.66 | 97.44 | 42.3 | 31.3 | 269.65 | 474.69 | 1.36× |
| | DiffRate | 85.38 | 97.39 | 38.5 | 37.5 | 249.39 | 513.26 | 1.47× |
| | ToaSt | 81.86 | 95.84 | 42.3 | 31.3 | 262.85 | 486.96 | 1.40× |
| | ToaSt | **88.94** | **97.95** | 38.5 | 37.5 | 242.88 | 527.01 | 1.51× |
| ViT-MAE-Huge | Baseline | 86.88 | 98.07 | 167.4 | – | 987.02 | 129.68 | – |
| | ToMe (r=7) | 85.31 | 97.67 | 92.9 | 44.5 | 564.04 | 226.94 | 1.75× |
| | ToMe (r=5) | 86.28 | 97.88 | 113.9 | 31.9 | 688.44 | 185.93 | 1.43× |
| | DiffRate | 86.65 | 97.88 | 103.4 | 38.2 | 631.02 | 202.85 | 1.56× |
| | ToaSt | **90.03** | **98.77** | 103.4 | 38.2 | 631.11 | 202.82 | 1.56× |
| | ToaSt | 88.52 | 98.29 | 101.4 | 39.4 | 620.73 | 206.21 | 1.59× |
| | ToaSt | 86.92 | 97.68 | 100.4 | 40.0 | 622.45 | 205.64 | 1.59× |

*Table 24.* Detailed results for Swin Transformer models on ImageNet-1K (H100 GPU, batch size 128).

| Model | Method | Top-1 (%) | Top-5 (%) | GFLOPs | FLOPs ↓ (%) | Latency (ms) | Throughput (img/s) | Speedup |
|---|---|---|---|---|---|---|---|---|
| Swin-Tiny | Baseline | 81.20 | 95.50 | 4.5 | – | 49.02 | 2610.94 | – |
| | ToaSt | **81.76** | **95.70** | 3.1 | 31.3 | 45.58 | 2705.76 | 1.04× |
| Swin-Small | Baseline | 83.20 | 96.20 | 8.7 | – | 83.42 | 1534.41 | – |
| | STViT-R | 82.60 | 96.07 | 5.8 | 33.3 | 77.74 | 1646.58 | 1.07× |
| | ToaSt | **84.65** | **96.80** | 5.4 | 38.2 | 67.03 | 1909.46 | 1.24× |
| Swin-Base | Baseline | 83.50 | 96.50 | 15.4 | – | 116.35 | 1100.10 | – |
| | STViT-R | 83.20 | 96.40 | 10.3 | 33.1 | 106.12 | 1206.23 | 1.10× |
| | ToaSt | **85.21** | 96.50 | 8.8 | 42.7 | 90.87 | 1408.60 | 1.28× |

*Table 25.* Block-wise latency profiling on H100 GPU (batch size 128, fp32). ToMe's bipartite matching overhead dominates in smaller models, while ToaSt maintains low overhead across all scales.

| Model | Method | Total (ms) | Attn (ms) | Selection (ms) | FFN (ms) | Overhead |
|---|---|---|---|---|---|---|
| DeiT Tiny | ToMe | 125.81 | 50.90 | 38.93 | 35.98 | 30.9% |
| | **ToaSt** | 53.01 | 29.42 | 8.79 | 14.80 | 16.6% |
| DeiT Small | ToMe | 37.90 | 13.80 | 10.56 | 13.54 | 27.9% |
| | **ToaSt** | 27.45 | 9.92 | 2.57 | 14.97 | 9.4% |
| DeiT Base | ToMe | 82.80 | 34.12 | 5.31 | 43.37 | 6.4% |
| | **ToaSt** | 75.59 | 23.18 | 3.51 | 48.91 | 4.6% |

## I. Detailed Experimental Configurations for Downstream task

*Table 26.* Detailed compression configurations for COCO object detection experiments.

| Backbone | MHSA Pruning | TCS Configuration |
|---|---|---|
| Swin-Small | 60% | FC1: 30% last 1L, FC2: 90% last 3L |
| Swin-Base (Config 1) | 60% | FC1: 30% last 1L, FC2: 90% last 4L |
| Swin-Base (Config 2) | 60% | FC1: 30% last 1L, FC2: 90% last 6L |

