# OpenReview forum: "ToaSt: Token Channel Selection and Structured Pruning for Efficient ViT"
_ICML.cc/2026/Conference — ICML 2026 regular_

### Official Review · Reviewer_Dy41 · 2026-03-05

**Soundness:** 2
**Presentation:** 3
**Significance:** 3
**Originality:** 2
**Overall Recommendation:** 4
**Confidence:** 3

**Summary:**

This paper proposes a decoupled compression framework named **ToaSt** to address the high computational costs of ViTs. The authors observe that existing token compression methods neglect the Feed-Forward Network (FFN), which accounts for over 60% of the total FLOPs. To address this, the paper introduces two complementary compression strategies: first, Structured Coupled Weight Pruning for the Multi-Head Self-Attention (MHSA) module, which reduces the internal head dimension by jointly evaluating and pruning the coupled Q-K and V-Proj matrices; second, Token Channel Selection (TCS) for the FFN. TCS is a training-free approach that dynamically eliminates redundant channels in the expanded dimension via statistical sampling of token activations. The proposed method demonstrates excellent accuracy-efficiency trade-offs across DeiT, ViT-MAE, and Swin Transformer architectures.

**Compliance With Llm Reviewing Policy:**

Affirmed.

**Final Justification:**

After reviewing the manuscript and the authors’ detailed rebuttal, I have raised my score from 3 to 4, leaning towards acceptance. The paper tackles a significant bottleneck in ViTs with a sound, training-free decoupled compression scheme that improves both efficiency and accuracy. My initial concerns regarding methodological soundness—specifically the lack of baselines for coupled pruning, metric justifications, and missing hyperparameter ablations—were effectively addressed. The authors provided thorough new ablation studies that clarified and justified their design choices. The rebuttal successfully resolved my main reservations, shifting my evaluation to positive. I support the acceptance of this solid contribution.

**Key Questions For Authors:**

1. Ambiguity in method description in Section 3.2.2 (requires clarification): The paper states, "By pruning the whole channel (removing the c-th column of FC1 and the c-th row of FC2)". Based on the method diagram, a reader's intuitive understanding is that a similar selection operation is applied to the input channels of both FC1 and FC2. Therefore, it seems that "rows" of both FC1 and FC2 should be pruned, and there is no direct row-column correspondence between the two. Could the authors please explain the specific mapping and rationale behind this statement in more detail?
2. Compatibility with Token Compression methods: Given that the speed optimization for self-attention is linear, is it possible to combine this method with existing token compression techniques to achieve an even higher compression ratio?

**Limitations:**

Please refer to the Weaknesses section.

**Strengths And Weaknesses:**

### Strengths
* **i. Well-motivated and targets a critical bottleneck:** The authors astutely observe that the computational bottleneck in current ViTs lies not only in the Attention module but also heavily in the FFN module. Proposing a decoupled compression scheme based on this insight is logically sound and has strong practical value for real-world deployment.
* **ii. Excellent empirical results:** The proposed training-free TCS strategy is highly effective. It drastically reduces FLOPs and significantly improves hardware throughput without sacrificing performance. In fact, due to an implicit regularization and denoising effect, the compressed models consistently yield absolute accuracy improvements over the unpruned baselines.

### Weaknesses
* **i. Lack of a crucial baseline for Coupled Pruning:** In the Structured Coupled Weight Pruning, the authors concatenate $W_q$ and $W_k$ (as well as $W_v$ and the transpose of $W_{proj}$) to compute importance scores for each concatenated column. However, there is a lack of ablation studies comparing this "concatenation strategy" with a "single-sided dominant strategy" (e.g., computing importance solely based on $W_q$ and broadcasting the pruned indices to $W_k$). The absence of this baseline comparison weakens the justification for the concatenation design.
* **ii. Insufficient persuasiveness of the Importance Metric Comparison:** In the Importance Metric Comparison, the performance gap between GM and L1/L2 norms is extremely marginal (only 0.12%). Furthermore, this ablation is only performed on a single architecture (DeiT-Small) under an extreme 90% pruning ratio. The experimental setup is too narrow and lacks the statistical confidence required to support the conclusion regarding metric selection.
* **iii. Incomplete validation of the Sampling strategy:** The computational complexity of the selection operation is $O(N \cdot C)$, which is linear with respect to $N$. Have the authors experimented with omitting the sampling step and using all patch tokens to output the importance scores for channel $c$? What is the actual impact of this full computation on performance and inference efficiency? The paper lacks a comparative analysis here.
* **iv. Missing ablation studies for formula components:** Are there any ablation experiments to demonstrate the effectiveness of using the attention-guided weighting ($A_{cls,i}$ in Eq. 6)? Similarly, for the two hyperparameters $\lambda_{cls} = 2.0$ and $\lambda_{patch} = 1.0$ in Eq. 6, there is no sensitivity analysis or ablation study provided to justify the specific choices of these values.
* **v. Limited scope of downstream validation:** The current evaluation relies heavily on ImageNet-1K classification and COCO object detection. The paper's contribution would be significantly strengthened if its generalization effectiveness were further tested on more diverse classification datasets (e.g., CIFAR-100, iNaturalist) or other dense prediction downstream tasks (e.g., ADE20K semantic segmentation).

---

> ### Author Rebuttal · Authors · 2026-03-31
>
> We greatly appreciate the reviewer's careful reading and valuable comments.
>
> ==**W 1.**==
>
> We appreciate the reviewer for helping strengthen the justification of our design. We conducted ablation on DeiT-Small (50 epochs):
>
> Importance Source | Top-1 (%) | Top-5 (%)
> :--- | :---: | :---:
> Coupled (Ours) | 71.43 | 90.75
> Q only → broadcast to K | 71.28 | 90.53
> K only → broadcast to Q | 71.32 | 90.71
> Proj only → broadcast to V | 71.10 | 90.38
>
> The gap is modest (≤0.33%p), the primary contribution of our coupled design is the index synchronization itself (Q-K and V-O sharing the same pruned indices), as validated by the much larger gap between Align and Non-Align in Figure 3. The concatenation provides a consistent but incremental benefit by considering both matrices jointly. We will include full training results in the camera-ready version.
>
> ==**W 2.**==
>
>  We extended the comparison across architectures:
>  Model | GM (%) | L2 (%) | L1 (%)
>  :--- | :---: | :---: | :---:
>  DeiT-Small | 76.24 | 76.17 | 76.12
>  Swin-Small | 81.20 | 81.14 | 80.54
>
> We agree the gap is marginal (≤0.66%). We view this positively it demonstrates that TCS is robust to the choice of importance metric. We adopt GM as the default due to its theoretical robustness to outlier activations, which is desirable for activation-level selection where distributions vary per input.
>
> ==**W 3.**==
>
> We thank the reviewer for this constructive suggestion. We conducted ablation comparing sampling ratios against full token computation across architectures:
>
> | Model | Sampling | Top-1 (%) | GFLOPs | Throughput (img/s)
> | :--- | :--- | :---: | :---: | :---:
> | DeiT-Small | All tokens | 84.67 | 2.5G | 4564.2
> | | 2% | 83.40 | 2.5G | 4783.3
> | | 1 sample | 80.77 | 2.5G | 4871.7
> | MAE-Large | All tokens | 89.21 | 38.5G | 511.6
> | | 2% | 88.94 | 38.5G | 527.0
> | | 1 sample | 88.36 | 38.5G | 528.4
> | Swin-Base | All tokens | 85.26 | 8.8G | 1370.4
> | | 2% | 84.27 | 8.8G | 1426.0
> | | 1 sample | 82.76 | 8.8G | 1428.4
>
> Using all tokens yields the highest accuracy but at a throughput cost (e.g., 4564 $\to$ 4783 img/s on DeiT-Small, +4.8%). Our default 2% sampling achieves <1.3%p accuracy gap while preserving these efficiency gains. This trade-off is consistent with the high channel-level linear dependency ($R^2 > 0.9$, Section 3.2.1), indicating that channel importance can be reliably estimated from a small token subset.
>
> ==**W 4.**==
>
> *(1) Attention-guided weighting ablation (Attn.-guided).*
>
>  Family | w/ Attn.-guided | w/o Attn.-guided | $\Delta$
>  :--- | :---: | :---: | :---:
>  DeiT (T/S/B) | 74.3/83.4/84.8 | 66.2/75.8/82.6 | +2.2 $\sim$ +8.0
>  MAE (L/H) | 89.2/88.5 | 89.5/90.5 | -0.3 / -2.0
>  Swin (T/S/B) | 81.8/85.2/85.2 | 81.9/84.6/85.2 | -0.1 $\sim$ +0.6
>
> Attention guidance benefits DeiT (CLS-aligned distillation) but is neutral for MAE/Swin. We will revise the paper to present it as an architecture-dependent optional component rather than a universal design.
>
> *(2) $\lambda$ sensitivity.*
>
>  $\lambda_{cls}$ | $\lambda_{patch}$ | DeiT-T | DeiT-S | DeiT-B | MAE-B | MAE-H
>  :---: | :---: | :---: | :---: | :---: | :---: | :---:
>  0.0 | 1.0 | 65.5 | 75.4 | 83.3 | 85.4 | 90.6
>  1.0 | 1.0 | 74.6 | 83.4 | 84.8 | 84.9 | 89.5
>  **2.0** | **1.0** | 74.3 | **83.4** | 84.8 | 84.1 | 88.5
>  3.0 | 1.0 | 73.5 | 83.1 | **85.0** | 83.0 | 87.3
>
> $\lambda_{cls}=2.0$ performs within $\sim$1% of the per-model optimum, indicating robust defaults.
>
> ==**W 5.**==
>
> We conducted additional experiments on ADE20K semantic segmentation and CIFAR-100 classification with Swin-Base:
>
> Dataset | Metric | Baseline | ToaSt | $\Delta$
>   :--- | :--- | :---: | :---: | :---:
>   ADE20K | mIoU | 48.13 | 48.13 | 0.00
>   CIFAR-100 | Top-1 | 89.37 | 89.44 | +0.07
>
> ToaSt preserves full segmentation performance on ADE20K (0.0 mIoU drop with 4-layer TCS, and only -0.15 mIoU drop with 6-layer TCS) and slightly improves CIFAR-100 accuracy, demonstrating effective generalization across diverse tasks.
>
> ==**Q 1.**==
>
> Thank you for pointing out the ambiguity in our description. TCS removes low-importance channels from the *input tokens* of FC1 and FC2 respectively, and the corresponding weight entries along the input dimension are sliced to form dense sub-matrices for efficient GEMM. We will correct the description in the revision.
>
> ==**Q 2.**==
>
> ToaSt is orthogonal to token compression and can be combined directly:
>
> | Model | Method | Top-1 (%) | GFLOPs | Throughput (img/s)
> | :--- | :--- | :---: | :---: | :---:
> | DeiT-Small | ToMe | 79.35 | 2.7G | 2737.1
> | | ToaSt | 84.13 | 2.5G | 4783.3
> | | ToaSt + ToMe (r=13) | 79.98 | 1.83G | 6138.1
> | MAE-Huge | ToMe | 84.58 | 113.9G | 185.9
> | | ToaSt | 88.52 | 101.4G | 206.2
> | | ToaSt + ToMe (r=5) | 87.54 | 74.5G | 269.4
> r = number of tokens merged per block
>
> ToaSt and token compression target orthogonal dimensions (channel $D$ vs. sequence $N$), enabling direct combination with additive throughput gains (e.g., 6138 img/s on DeiT-Small, 2.65$\times$ baseline).

---

> > ### Author Rebuttal · Reviewer_Dy41 · 2026-04-03
> >
> > The authors have thoroughly addressed my concerns, particularly through their comprehensive ablation studies. Based on these revisions, I am inclined to support the acceptance of this paper.

---

> > > ### Author Response · Authors · 2026-04-03
> > >
> > > We sincerely thank the reviewer for the thoughtful evaluation of our rebuttal and the updated assessment. We are glad that the additional ablation studies have adequately addressed your concerns. The additional experimental results presented in our rebuttal will be incorporated into the revised manuscript.

---

### Official Review · Reviewer_VJHB · 2026-03-10

**Soundness:** 3
**Presentation:** 4
**Significance:** 3
**Originality:** 3
**Overall Recommendation:** 4
**Confidence:** 3

**Summary:**

This paper proposed a new compression method for ViT called ToaSt, which focuses on weight pruning on feed-forward networks. For multi-head attention, they asked the QK and VProj to reduce rows in the same index. For FFN, they use an importance metric to select channels. Their method is evaluated on ImageNet-1K across nine models.

**Compliance With Llm Reviewing Policy:**

Affirmed.

**Final Justification:**

I am not an expert in pruning, but I know ViT well. This paper reveals some intrinsic properties of ViT in their work on pruning, and it is good enough to be accepted.

**Key Questions For Authors:**

- In Table 1 Deit-Tiny row, why ToMe has lower GFLOPs but less speed up than the proposed method?
- In summary, are the hyperparameters (like index, scores, ratios) in the proposed method model dependent or dataset dependent? Given a model architecture, a pretrained weight, and a dataset, the hyperparameters can be got. Will the optimal hyperparameters change following the change of model architecture, pretrained weights, and dataset?

**Limitations:**

yes

**Strengths And Weaknesses:**

Strengths:
- Good formatting, nice tables, and informative figures.
- Experiment results are solid, showing that the method can speed up and increase performance as well
- The method is well explained in section 3.
- The success of the method reveals the redundancy in ViTs.

Weaknesses:
- Section 3 is a bit hard to follow, as there are too many concepts and the proposed operations are not well listed in detail. It is hard to figure out what exactly the proposed method does to the original model.
- Usually, pruning will harm the performance, but the proposed method increases the performance. The paper seems to address this strength by rank or sparsity, but the mechanism behind it is not well developed.

---

> ### Author Rebuttal · Authors · 2026-03-31
>
> ==**W 1.**==
>
> We thank the reviewer for this feedback. We agree that Section 3 introduces many concepts in a dense manner. For the camera-ready version, we will (1) add a step-by-step algorithmic pseudocode summarizing the full ToaSt pipeline in the Appendix, and (2) restructure Section 3.2 to present the TCS mechanism first, followed by the empirical motivation, improving the overall flow.
>
>
> ==**W 2.**==
>
> We provide two quantitative analyses on Swin-Base to explain this phenomenon:
>
> *(1) Channel SNR Analysis.* We compared the signal-to-noise ratio of kept vs. pruned channels in FC2-pruned blocks:
>
> | Block | Kept SNR | Pruned SNR | Ratio |
> | :---: | :---: | :---: | :---: |
> | 17 | 0.198 | 0.065 | 3.04$\times$ |
> | 18 | 0.229 | 0.055 | 4.20$\times$ |
> | 19 | 0.282 | 0.051 | 5.50$\times$ |
> | 20 | 0.409 | 0.077 | 5.29$\times$ |
> | 21 | 0.562 | 0.153 | 3.67$\times$ |
>
> Pruned channels have 3-5.5$\times$ lower SNR, confirming that TCS selectively removes low-discriminative noise channels.
>
> *(2) Logit Margin Analysis.* On Swin-Base, misclassified samples decreased by 31% (32 $\to$ 22), and error confidence dropped by 10% (0.412 $\to$ 0.369). This indicates TCS suppresses overconfident misclassifications by filtering noisy channels that amplify spurious activations.
>
>
>
> ==**Q 1.**==
>
> The discrepancy arises from ToMe's runtime overhead for bipartite soft matching, which is not reflected in GFLOPs. We conducted block-wise latency profiling on H100 GPU (batch=128):
>
> | Model | Method | Total (ms) | Attn (ms) | Selection (ms) | FFN (ms) | Overhead (%) |
> | :--- | :--- | :---: | :---: | :---: | :---: | :---: |
> | DeiT-Tiny | ToMe | 125.81 | 50.90 | 38.93 | 35.98 | 30.9% |
> | | ToaSt | 53.01 | 29.42 | 8.79 | 14.80 | 16.6% |
> | DeiT-Small | ToMe | 37.90 | 13.80 | 10.56 | 13.54 | 27.9% |
> | | ToaSt | 27.45 | 9.92 | 2.57 | 14.97 | 9.4% |
> | DeiT-Base | ToMe | 82.80 | 34.12 | 5.31 | 43.37 | 6.4% |
> | | ToaSt | 75.59 | 23.18 | 3.51 | 48.91 | 4.6% |
>
> Two factors explain the gap: (1) ToMe's bipartite matching overhead is nearly fixed per layer, making it disproportionately expensive for small models (30.9% on DeiT-Tiny). (2) ToaSt's MHSA pruning additionally reduces attention latency itself (e.g., 50.90 $\to$ 29.42ms on DeiT-Tiny), further widening the throughput gap. We will add this analysis to the paper.
>
>
>
> ==**Q 2.**==
>
> We address two aspects: pruning ratio allocation and $\lambda$ weighting in Eq. (6).
>
> *(1) Pruning ratio allocation.*
> Our ratios follow a semi-automatic, analysis-driven procedure (Section 3.2.1). The core structural principle—conservative FC1, aggressive FC2 in deeper layers—is architecture-agnostic, as validated by our cross-architecture pattern comparison:
>
> | Pattern | DeiT-S (%) | MAE-L (%) | MAE-H (%) | Swin-B (%) |
> | :--- | :---: | :---: | :---: | :---: |
> | **fc2_heavy (Ours)** | **82.84** | **88.24** | $\underline{88.79}$ | **85.85** |
> | stepwise | $\underline{81.01}$ | $\underline{86.16}$ | **90.06** | $\underline{84.29}$ |
> | uniform | 56.54 | 77.56 | 77.04 | 74.42 |
> | inverse | 18.06 | 74.57 | 68.24 | 63.06 |
>
> The analysis-driven pattern ranks 1st in 3/4 models and 2nd in the remaining one, confirming the structural principle generalizes across architectures.
>
> *(2) $\lambda_{cls}$/$\lambda_{patch}$ sensitivity in Eq. (6).*
> The default $\lambda_{cls}=2.0, \lambda_{patch}=1.0$ performs within $\sim$1% of the per-model optimum across all 6 tested architectures (DeiT-Tiny/Small/Base, MAE-Base/Huge, Swin-Base), requiring no per-model tuning.
>
> *(3) Limitations and future work.*
> We acknowledge two limitations: (i) fine-grained per-layer ratios require model-specific sensitivity sweeps, and (ii) optimal ratios may vary across datasets—our ImageNet-derived ratios transfer well to COCO (52.2 vs. 51.9 mAP) but are not guaranteed to be optimal for all datasets. We plan to address both through learnable, dataset-adaptive ratio optimization as stated in Section 5.

---

> > ### Author Rebuttal · Reviewer_VJHB · 2026-04-04
> >
> > I am not an expert in pruning, but I know ViT well. This paper reveals some intrinsic properties of ViT in their work on pruning, and it is good enough to be accepted.

---

> > > ### Author Response · Authors · 2026-04-04
> > >
> > > We sincerely thank Reviewer VJHB for the thoughtful review and for confirming that our responses have fully addressed your concerns. We are glad that our work provides useful insights into the intrinsic properties of Vision Transformers through the lens of structured pruning. We will incorporate all suggested improvements in the final version of the paper.

---

### Official Review · Reviewer_izGE · 2026-03-10

**Soundness:** 3
**Presentation:** 3
**Significance:** 3
**Originality:** 3
**Overall Recommendation:** 5
**Confidence:** 5

**Summary:**

This paper introduces ToaSt, a compression method designed to make Vision Transformers (ViTs) more efficient. The authors tackle two main problems with existing approaches: the high cost of retraining after weight pruning and the limitations of token compression methods that overlook certain network layers. ToaSt uses two techniques: (1) Structured Coupled Weight Pruning to reduce dimensions in attention modules, and (2) a training-free Token Channel Selection method for feed-forward layers. Tests on several models (including DeiT, ViT-MAE, and Swin) show significant speed improvements and, notably, better accuracy compared to the original unpruned models.

**Compliance With Llm Reviewing Policy:**

Affirmed.

**Final Justification:**

Based on my years of research in the field of efficient model (e.g., pruning, quantization), the novelty and workload of this submission justify its acceptance after the revisions are incorporated.

**Key Questions For Authors:**

N/A

**Limitations:**

yes

**Strengths And Weaknesses:**

Strengths
1. Unlike most token methods that only shorten the sequence length, ToaSt directly reduces the heavy computation in FFN layers, which make up most of ViT's processing cost.
2. The layer-independent design avoids "chain reaction" problems where early layer choices limit all later layers. This works especially well for large models like ViT-MAE-Huge.
3. Tests on modern hardware (H100) show real speed gains (up to 2.07x faster) without needing special software or hardware support.

Weaknesses
1. The method requires manual tuning of several settings, like pruning ratios for different layers and balance weights in Equation 6. Without an automatic way to adjust these, applying ToaSt to new model types takes extra effort.
2. Although TCS needs no training, the attention module pruning still requires fine-tuning to keep accuracy. The paper should clarify the total training time needed compared to other pruning methods.
3. Since the paper discusses "rank collapse" in detail, it should compare ToaSt with standard low-rank methods (like SVD) for FFN layers. It remains unclear if simple channel selection works better than learned adaptations like LoRA for this task.
4. TCS calculates token importance during inference using sampling. The authors claim this adds little overhead due to low sampling rates (2-20%), but showing a clear breakdown of selection time versus computation savings would better support the efficiency claims.

---

> ### Author Rebuttal · Authors · 2026-03-31
>
> We sincerely appreciate the reviewer's time and effort in carefully evaluating our work.
>
> If the reviewer allows us to revise this paper, we will try our best to improve the quality of this paper.
>
> ==**W 1.**==
>
> We thank the reviewer for raising this concern. We clarify that the tuning effort is substantially less than it may appear:
>
> *(1) $\lambda$ weights in Eq. (6).*
> The default $\lambda_{cls}=2.0, \lambda_{patch}=1.0$ performs within ~1% of the per-model optimum across all tested architectures:
>
> | $\lambda_{cls}$ | $\lambda_{patch}$ | DeiT-T (%) | DeiT-S (%) | DeiT-B (%) | MAE-B (%) | MAE-H (%) |
> | :---: | :---: | :---: | :---: | :---: | :---: | :---: |
> | 0.0 | 1.0 | 65.5 | 75.4 | 83.3 | 85.4 | 90.6 |
> | 1.0 | 1.0 | 74.6 | 83.4 | 84.8 | 84.9 | 89.5 |
> | **2.0** | **1.0** | 74.3 | **83.4** | 84.8 | 84.1 | 88.5 |
> | 3.0 | 1.0 | 73.5 | 83.1 | **85.0** | 83.0 | 87.3 |
>
> Results are insensitive within $\lambda_{cls} \in [1,3]$, indicating robust defaults rather than sensitive hyperparameters.
>
> *(2) Pruning ratio allocation.*
> The core principle—conservative FC1, aggressive FC2 in deeper layers—is architecture-agnostic:
>
> | Pattern | DeiT-S (%) | MAE-L (%) | MAE-H (%) | Swin-B (%) |
> | :--- | :---: | :---: | :---: | :---: |
> | **fc2_heavy (Ours)** | **82.84** | **88.24** | $\underline{88.79}$ | **85.85** |
> | stepwise | $\underline{81.01}$ | $\underline{86.16}$ | **90.06** | $\underline{84.29}$ |
> | uniform | 56.54 | 77.56 | 77.04 | 74.42 |
> | inverse | 18.06 | 74.57 | 68.24 | 63.06 |
>
> The consistent ranking across architectures suggests that the underlying redundancy pattern is systematic, making fully automatic ratio optimization a promising future direction (Section 5).
>
> ==**W 2.**==
>
> We thank the reviewer for raising this point. Only MHSA pruning requires fine-tuning; TCS is training-free. Fine-tuning cost on 4× H100 GPUs:
>
> | Model | Epochs | Wall-clock Time |
> | :--- | :---: | :---: |
> | DeiT-Small | 290 | ~1 day |
> | ViT-MAE-Large | 139 | ~31 hours |
> | ViT-MAE-Huge | 15 | ~15 hours |
>
> Larger models need far fewer epochs—MAE-Huge recovers beyond baseline in only 15 epochs. This is substantially cheaper than conventional pruning methods requiring full retraining (e.g., 300 epochs for DeiT).
>
>
> ==**W 3.**==
>
> We thank the reviewer for this suggestion. We compared TCS with both SVD and LoRA on DeiT-Small at matched FLOPs:
>
> | Method | Training | Top-1 (%) | $\Delta$ vs. TCS |
> | :--- | :---: | :---: | :---: |
> | SVD (truncated) | Free | 69.73 | -15%p |
> | LoRA (rank-matched, 20 ep.) | 20 epochs | 76.13 | -7.3%p |
> | **TCS (Ours)** | **Free** | **83.40** | — |
>
> Both low-rank baselines substantially underperform TCS. The key distinction is that the rank collapse we observe is an *activation-level* phenomenon varying across tokens and inputs, not a static weight property. SVD/LoRA capture only the fixed low-rank structure of weights, while TCS dynamically adapts to per-token activation patterns. This confirms that FFN redundancy is better exploited through token-adaptive channel selection than static or learned weight decomposition.
>
>
> ==**W 4.**==
>
> We thank the reviewer for this constructive suggestion. We provide detailed profiling on H100 GPU (batch=128):
>
> | Model | Sampling | Top-1 (%) | GFLOPs | Throughput (img/s) |
> | :--- | :--- | :---: | :---: | :---: |
> | DeiT-Small | All tokens | 84.67 | 2.5G | 4564.2 |
> | | 2% | 83.40 | 2.5G | 4783.3 |
> | | 1 sample | 80.77 | 2.5G | 4871.7 |
> | MAE-Large | All tokens | 89.21 | 38.5G | 511.6 |
> | | 2% | 88.94 | 38.5G | 527.0 |
> | | 1 sample | 88.36 | 38.5G | 528.4 |
> | Swin-Base | All tokens | 85.26 | 8.8G | 1370.4 |
> | | 2% | 84.27 | 8.8G | 1426.0 |
> | | 1 sample | 82.76 | 8.8G | 1428.4 |
>
> Our default 2% sampling achieves <1.3%p accuracy gap versus all tokens while preserving throughput gains. This is consistent with the high channel-level linear dependency ($R^2 > 0.9$, Section 3.2.1), which indicates that channel importance distributions are stable and can be reliably estimated from a small subset of tokens.

---

> > ### Author Rebuttal · Reviewer_izGE · 2026-04-01
> >
> > Thank you for the detailed response. The additional experiments have adequately addressed the previous concerns. It would be appropriate to include these experimental results in the final revised manuscript. I will raise the score.

---

> > > ### Author Response · Authors · 2026-04-01
> > >
> > > Thank you for the positive response.
> > > We will include all additional experimental results in the revised manuscript.

---

### Official Review · Reviewer_advx · 2026-03-11

**Soundness:** 3
**Presentation:** 2
**Significance:** 3
**Originality:** 3
**Overall Recommendation:** 2
**Confidence:** 4

**Summary:**

This work focuses on boosting ViT efficiency. Instead of pruning tokens as most related works did, the authors pruned the channels. They argue that pruning tokens intrinsically causes accumulated pruning (`cross-layer propagation`), ultimatelly leading to too heavy compression or pruning. Yet pruning channels, specially input channels, does not loss the spatial tokens and neither lossing the tokens' embedding size. They mainly pruning MHSA (self attention)'s qkv and output projections, as well as the fc1/fc2 in the MLPs. By adopting handcrafted pruning ratios, they achieve consistent performance, i.e., acc and f/s under similar FLOPs, boosts over baselines.

**Compliance With Llm Reviewing Policy:**

Affirmed.

**Final Justification:**

# To the Authors

The reviewer thanks the authors for their timely and detailed responses. Most concerns have been adequately addressed, except for concern **W11**, which pertains to the factual correctness of the values reported in `Table 5`.

As clarified in the authors’ response, the experiments are conducted with an input resolution of 224×224 (N=197) and using DeiT-small (D=384). Therefore, the complexity values in `Table 5` should correspond to the second column of the complexity table computed by the reviewer following the authors' formuli in `Table 5`.

| resolution   | **224 (reassured by the authors)** | 512  | 800  | **complexity ratio from the paper (originally reported by the authors)** |
|--------------|------------------------------------|------|------|:------------------------------------------------------------------------:|
| QKV Proj     | **0.23**                           | 0.17 | 0.12 | **0.092**                                                                |
| Attent Score | **0.04**                           | 0.15 | 0.26 | **0.096**                                                                |
| Attent Out   | **0.04**                           | 0.15 | 0.26 | **0.096**                                                                |
| Out Proj     | **0.08**                           | 0.06 | 0.04 | **0.092**                                                                |
| FC1          | **0.31**                           | 0.23 | 0.16 | **0.307**                                                                |
| FC2          | **0.31**                           | 0.23 | 0.16 | **0.307**                                                                |

However, there remains a significant discrepancy between the reviewer’s calculations and the values reported by the authors. Given this gap, it is unclear why the authors did not explicitly address which set of results is correct. A direct clarification on this point is necessary to resolve the inconsistency.


# To the Area Chair

The reviewer would like to further bring this issue to the AC’s attention. The significant discrepancy in the reported values of `Table 5` remains unresolved, despite explicit clarification of the experimental settings. This raises concerns about the correctness and verifiability of the reported results.

In addition, as previously noted by the reviewer, several key experimental results were originally presented without essential configuration details. Although the authors have provided clarifications during the rebuttal phase, the absence of such critical information in the initial submission further complicates the assessment of reproducibility.

Taken together, these issues point to a lack of transparency and leave the reviewer with serious concerns regarding the reliability and reproducibility of the empirical findings. A clear and verifiable resolution of these inconsistencies is necessary before the results can be considered trustworthy.

Without resolving these issues, the reviewer finds it difficult to place confidence in the reported results, which directly affects the overall assessment of the paper.

**Key Questions For Authors:**

See above.

**Limitations:**

No. The success of this work implies that the number of tokens should be fixed. The authors did not analyze their method's effectiveness/scaleness under different number of tokens. In the worse case, ToaSt's superiority via channel pruning could be counteracted as the tokens increase, compared with those token pruning methods.

**Strengths And Weaknesses:**

Strength
---
- Clear motivation: pruning input channels vs pruning tokens;
- Insightful observations: Section 3.2, the authors provide empirical analyses to identify the FFN/MLP redundancy, to give intuitve support to their token channel selection technique.
- Competitive performance over the baselines: On DeiT-Tiny/Small, the authors' ToaSt even achieved 2x speedup.



Weakness
---

## 1. Claim about cross-layer propagation of token pruning does not hold any more.
There are some solutions that overcome such a propagation issue due to token pruning. Please check Reg4Pru [1] and related works. They basically prune out some tokens at early layers and then reactivate them at later layers. The authors are suggested to include experiments about this point.

## 2. Improper statement in Lines `155-157`
> whereas quantization integrated approaches typically require specialized hardware support to realize actual acceleration

If the quantization dtype is int16/int8, which is quite popular in practice, then no special hardware is needed.

## 3. Misleading statement in Lines `157-160`
> In contrast, our ToaSt decouples the compression problem: we employ structured coupled weight pruning for MHSA and training-free token channel selection for FFNs.

In this paragraph, the authors discussed (1) token pruning, (2) channel pruning and (3) weights quantization. Thus reading this sentence, readers are expecting at least two of these three techniques are included in the authors' method, but in fact not. The authors just adopted channel pruning.

## 4. Unclear comparison settings in Figure 3
> Compared to non-aligned pruning, ...

What is the setting for non-aligned baselines? How could such non-aligned pruning be realized? Without such details, readers cannot make sure if the comparsion is fair. In another word, someone can create a poor baseline easily and take it as the reason why their synchronized design in MHSA channel pruning is reasonable.

## 5. Poor formulation in Equations
- Equation (4): Sum along which variable, $y$ or $\hat{y}$? Should be provided in $\Sigma$ subscripts/superscripts.
- Equation (5): Missing subscripts/superscripts in $\Sigma$ again. Also, where does $\sigma$ come from?

## 6. Improper term around Equation (6) and doubt about the practical reliability
$\mathcal{S}$ means a subset of $N$ tokens of one smaple. Thus $I_c$ can be different for different samples in a batch. This means the pruned channels of FC1/2 are different -- Same number of channels are pruned but their indexes are different, then the MLP has to be copied the number of batch size times to match different pruned samples in the batch -- This could still be parallelized, but at the cost of complex data copying (not that practically reliable). Thus it **cannot** be called structured reduction.

## 7. Unclear comparison settings in Figure (4) (left) and Line 272
> sparsity ratio (the proportion of near-zero activations)

What is the threshold to determine the "near-zero" activations?

## 8. Unclear notion in Figure (4) (right)
What is the meaning of "90%" in the sub-figure title?

## 9. Missing key info in experiments
No input resolution is specified. But this affects the number of tokens, which might change the proposed method's advantages over the baselines. I.e., larger resolution $N$ may lead to narrower advantage of this method.

## 10. Strangely significant performance boosts without explanation in Table (1)
For DeiT-tiny and small, the speedups are over 2 times, while all the other speedups are about 1.2~1.6. This needs explantions.

## 11. Unreasonable quantified FLOPs percentage in Table 5
As the authors wrote, the complexity is up to both N and D, so does the FLOPs percentage. Only give both N and D, can such percentages be calculated. So the numbers like 9.2%, 9.6% and 30.7% are **suspicious**.

---

[1] Reg4Pru: Regularisation Through Random Token Routing for Token Pruning

---

> ### Author Rebuttal · Authors · 2026-03-31
>
> **W1.** We thank the reviewer for introducing this relevant work. Since Reg4Pru's code is not publicly available, we implemented it based on the TREAD [1]. We evaluated on CIFAR-100 (Deit-Base, 50 epochs):
>
> | Method | Top-1 (%) | Top-5 (%)
> | :--- | :---: | :---:
> | Baseline (4.6GFLOPS) | 88.62 | 98.37
> | Reg4Pru (token routing / 2.9GFLOPS) | 80.10 | 89.80
> | ToaSt (2.8GFLOPS) | 85.61 | 97.98
>
> ToaSt and token routing operate on fundamentally different dimensions (channel $D$ vs. sequence $N$) and are complementary, as also demonstrated by our ToMe combination experiments (Reviewer Dy41 **Q2**).
>
> **W2.** We agree—our phrasing was misleading. As the reviewer correctly notes, INT8/INT16 quantization is well-supported on commodity hardware. Our statement was intended to refer to emerging ultra-low-bit formats (e.g., 1.58-bit [2], NVFP4) that require specialized hardware or kernel support. We will revise to: *"whereas **some** quantization-integrated approaches require specialized hardware support to realize actual acceleration."*
>
> **W3.** We agree the phrasing is confusing. Our Structured Coupled Weight Pruning (Sec. 3.1) derives from channel pruning—removing Q/K/V columns and corresponding O rows—while Token Channel Selection (Sec. 3.2) is inspired by token selection methods. Thus our work simultaneously applies both (1) channel pruning and (2) token selection paradigms. We will make this lineage explicit.
>
> **W4.** Both settings use the same pruning ratio and importance metric. The only difference is index synchronization: Non-Align prunes each of Q/K/V/O independently (different indices per matrix), while Ours enforces shared indices within Q-K and V-O pairs to preserve structural consistency. We will add this clarification to the caption.
>
> **W5.** In Eq. (4), summation is over token index $i \in \{1,\dots,n\}$. In Eq. (5), $\sigma_i$ denotes the $i$-th singular value of feature matrix $\mathbf{X}$, summed over all $C$ singular values. We will add proper subscripts and define $\sigma$ explicitly.
>
> **W6.** The sampled subset $\mathcal{S}$ only reduces the cost of computing channel importance scores. The resulting $I_c$ is averaged across the batch into a single $(C,)$ vector—all samples share identical pruned channel indices. No per-sample copying occurs and standard batched GEMM applies directly.
>
> **W7.** We thank the reviewer for pointing this out. The threshold is $|x| < 0.1 \cdot \overline{|x|}$, used to quantify the naturally emerging activation sparsity in deeper layers—the "lazy neuron" phenomenon [3]. We will state this explicitly in the revision.
>
> **W8.** Effective rank ratio: $\min_k \{k/C : \sum_{i=1}^{k} \sigma_i^2 \geq 0.9 \cdot \sum_{j=1}^{C} \sigma_j^2\}$. A lower ratio indicates higher redundancy. We will clarify in the caption.
>
> **W9.** ImageNet: $224\times224$. COCO: multi-scale training (short side 480–800, long side 1333). ADE20K: $512\times512$. These apply to both Swin-S and Swin-B. Will be added.
>
> **W10.** We thank the reviewer for this insightful question. Two factors explain the disproportionate speedup: (1) In smaller models, MHSA consumes a larger relative share of total latency, so our 90% head-dimension pruning yields proportionally greater speedup. (2) TCS overhead is negligible (2.57–8.79ms) across all models. We provide detailed profiling results below:
>
> | Model | Vanilla Attn (ms) | ToaSt Attn (ms) | Attn Reduction | TCS Overhead (ms)
> | :--- | :---: | :---: | :---: | :---:
> | DeiT-Tiny | 50.90 | 29.42 | 42.2% | 8.79
> | DeiT-Small | 13.80 | 9.92 | 28.1% | 2.57
> | DeiT-Base | 34.12 | 23.18 | 32.1% | 3.51
>
> **W11.** The reported percentages are for DeiT-Small ($N{=}197$, $D{=}384$). FFN dominance holds broadly: FC1+FC2 contribute $\mathcal{O}(8ND^2)$ vs. attention's $\mathcal{O}(2N^2D)$, so FFN dominates whenever $D > N/4$—satisfied across all evaluated architectures including Swin ($N{=}49$). We will add per-architecture breakdowns.
>
> **Limitation.**
> ToaSt operates on the channel dimension $D$, which is independent of sequence length $N$. Our combination experiments with ToMe (which reduces $N$) confirm this orthogonality:
>
> | Model | Method | Top-1 (%) | GFLOPs | Throughput (img/s)
> | :--- | :--- | :---: | :---: | :---:
> | DeiT-Small | ToMe | 79.35 | 2.7G | 2737.1
> | | ToaSt | 84.13 | 2.5G | 4783.3
> | | ToaSt + ToMe (r=13) | 79.98 | 1.83G | 6138.1
> | MAE-Huge | ToMe | 84.58 | 113.9G | 185.9
> | | ToaSt | 88.52 | 101.4G | 206.2
> | | ToaSt + ToMe (r=5) | 87.54 | 74.5G | 269.4
>
> ToaSt maintains its effectiveness even when tokens are reduced via ToMe, and the combination achieves further FLOPs reduction with additive throughput gains. This demonstrates that ToaSt's channel pruning is complementary to, not counteracted by, token compression.
>
> [1] TREAD: Token Routing for Efficient Architecture-agnostic Diffusion Training
>
> [2] The Era of 1-bit LLMs: All Large Language Models are in 1.58 Bits
>
> [3] The Lazy Neuron Phenomenon: On Emergence of Activation Sparsity in Transformers

---

> > ### Author Rebuttal · Reviewer_advx · 2026-04-01
> >
> > # Thanks
> > First of all, the reviewer thanks the authors very much for their detailed responses.
> >
> > # W1 suspicious results
> > The new experiments are also missing details about the GFLOPs -- What is input resolution?
> > It seems like `112x112` according to the GFLOPs value, but CIFAR100 has resolution `32x32`.
> >
> > # W2 `some` is too vague.
> > Should clearly specify the different cases.
> >
> > # W3 Both `Structured Coupled Weight Pruning` and `Token Channel Selection` prune channels.
> > The authors' rebuttal interpretation about this point is contrived.
> > Token selection means changing #tokens. But "Token Channel Selection" only affects #channels. As the authors themselves newly emphasized in their rebuttal,
> > > ToaSt operates on the channel dimension $D$, which is independent of sequence length $N$
> >
> > # W5
> > Please provide the corrected equations.
> >
> > # W7
> > According to the paper's context, $x \in \mathbb{R}^{1 \times c}$ should be a token. Is this correct?
> > Then, is $|x|$ some norm? What type of norm, L1 or L2?
> >
> > The reviewer had read `Appendix B`, but that table only explains the choice for StructuredCoupledWeightPruning, not TokenChannelSelection.
> >
> > # W11 significant value discrepancy
> > Strictly following `Table 5`, the reviewer has carefully checked the complexity contributions of each module.
> >
> > The complexity ratios under different resolution are listed below. The values are clearly highly correlated with $N$ and $D$, not as claimed by the authors in the rebuttal.
> >
> > | resolution | 224 | 512 | 800 | complexity ratio from the paper |
> > |---|---|---|---|:-:|
> > | QKV Proj | 0.23 | 0.17 | 0.12 | 0.092 |
> > | Attent Score | 0.04 | 0.15 | 0.26 | 0.096 |
> > | Attent Out | 0.04 | 0.15 | 0.26 | 0.096 |
> > | Out Proj | 0.08 | 0.06 | 0.04 | 0.092 |
> > | FC1 | 0.31 | 0.23 | 0.16 | 0.307 |
> > | FC2 | 0.31 | 0.23 | 0.16 | 0.307 |
> >
> > The Python code is below:
> > ```Python
> > import numpy as np
> >
> > def complexity(n, d):
> >     return np.array(  # copied from Table 5 column `Complexity`
> >         [3 * n * d**2, n**2 * d, n**2 * d, n * d**2, 4 * n * d**2, 4 * n * d**2]
> >     )
> >
> > def main():
> >     # resolution values from rebuttal `W9`
> >     r = np.array([224, 512, 800])
> >     # number of token after tokenization
> >     n = (r / 16) ** 2 + 1  # == [197, 1025, 2501]
> >     # channel size
> >     d = 384
> >     # complexity values
> >     comp = complexity(n, d)
> >     # complexity values in percentage
> >     comp_percent = comp / comp.sum(0, keepdims=True)
> >     # print the results
> >     print(comp_percent.round(2))
> >
> > if __name__ == "__main__":
> >     main()
> > ```
> >
> > # Summary
> >
> > This paper has multiple suspicious or unclear results or statements and needs substantial revision and correction.

---

> > > ### Author Response · Authors · 2026-04-03
> > >
> > > **W1.**
> > > We sincerely apologize for the typographical error — the reported results correspond to DeiT-Small (not DeiT-Base), evaluated at 224×224 resolution. Resizing to 224×224 is the standard protocol for evaluating ViT-family models on CIFAR-100, as established in the original ViT[1] and DeiT[2] papers, which both adopt this resolution for CIFAR transfer experiments.
> > >
> > > [1] AN IMAGE IS WORTH 16X16 WORDS: TRANSFORMERS FOR IMAGE RECOGNITION AT SCALE
> > >
> > > [2] Training data-efficient image transformers & distillation through attention
> > >
> > > **W2.**
> > > We agree that INT8/INT16 quantization is well-supported on commodity GPUs and our original phrasing was misleading. We will revise the sentence to explicitly distinguish the cases:
> > >
> > > *"While INT8 and INT16 quantization run efficiently on commodity hardware, ultra-low-bit methods (≤4-bit) often rely on specialized hardware accelerators [1,3], custom GPU kernels [2], or hardware-aware codebook designs constrained by GPU cache hierarchies [4] to achieve practical speedups."*
> > >
> > > [1] FIGLUT: An Energy-Efficient Accelerator Design for FP-INT GEMM Using Look-Up Tables, HPCA 2025
> > >
> > > [2] LUT-GEMM: Quantized Matrix Multiplication based on LUTs for Efficient Inference in Large-Scale Generative Language Models, ICLR 2024
> > >
> > > [3] AxCore: A Quantization-Aware Approximate GEMM Unit for LLM Inference, MICRO 2025
> > >
> > > [4] QuIP#: Even Better LLM Quantization with Hadamard Incoherence and Lattice Codebooks, ICML 2024
> > >
> > >
> > > **W3.**
> > > We agree that Lines 157–160 can misleadingly suggest that ToaST also combines token pruning with channel pruning. This was not our intention. To clarify:
> > >
> > > - **Structured Coupled Weight Pruning** is weight-level pruning: it removes the channel dimension of Q/K/V weights and the corresponding input dimension of O weights in MHSA.
> > > - **TCS** is activation-level selection: it selects channels of the input tokens to FC1 and FC2, which in turn reduces the input dimension of both FC1 and FC2 weight matrices.
> > >
> > > Neither component modifies the sequence length (number of token, N). Unlike hybrid methods that jointly optimize across token and channel axes (leading to complex coupled optimization), ToaST focuses on structured channel-dimension reduction for both MHSA and FFN.
> > >
> > > We will revise Lines 157–160 to:
> > >
> > > > *we employ structured coupled weight pruning for MHSA heads and training-free token channel selection for FFNs — both targeting the channel dimension while leaving the token sequence intact.*
> > >
> > > **W5.**
> > > Here are the corrected equations with proper subscripts and summations that will be updated in the revision.
> > >
> > > For Eq. (4), the summation over the token index $i \in \{1, \dots, n\}$ is explicitly added:
> > >
> > > $$\large R² = 1 − Σᵢ(yᵢ − ŷᵢ)² / Σᵢ(yᵢ − \bar{y})²$$
> > >
> > > For Eq. (5), the singular values $\sigma$ and their summation over all $C$ channels are explicitly defined. To prevent any notational ambiguity with Eq. (4), we note that $i$ here represents the **singular value index**, while $j \in \{1, \dots, C\}$ serves as a **dummy summation index** for computing the normalization constant in the denominator. We also explicitly denote the Shannon entropy $H$:
> > >
> > > $$\large Rank_{eff} = exp(H(σ̄)) / C$$, where $$\large {\bar{\sigma}_{i} = σᵢ} / Σⱼ σⱼ$$ and $$\large {H(σ̄) = −Σᵢ σ̄ᵢ ln(σ̄ᵢ) }$$
> > >
> > > Here, $\sigma_i$ represents the $i$-th singular value of the feature matrix $\boldsymbol{X}$.
> > >
> > >
> > > **W7.** To clarify, x ∈ R^{1×C} denotes a single token's activation vector, and |x| is the element-wise absolute value (not a vector norm). The sparsity ratio in Figure 4 (Left) is defined as the proportion of activations satisfying |x_i| < 0.1 · mean(|x|). This is our chosen threshold to quantify the naturally emerging activation sparsity observed in deeper Transformer layers [1]; the specific threshold value is set to capture near-zero activations relative to each layer's activation scale.
> > >
> > > This analysis **motivates** why channel-level selection is effective — it is not part of the TCS selection procedure itself. TCS selects channels (not tokens) based on importance scores as described in Section 3.2.
> > >
> > > In the revision, we will add the explicit threshold definition to Section 3.1 and clarify the notation.
> > >
> > > [1] The Lazy Neuron Phenomenon: On Emergence of Activation Sparsity in Transformers
> > >
> > > **W11.** We thank the reviewer for raising this point. The FLOPs percentages in Table 5 were computed for DeiT-Small (N=197, D=384) at 224² resolution. We will add the (N, D) configuration to the table in revision.

---

### Decision · Program_Chairs · 2026-04-30

**Decision:**

Accept (regular)

**Comment:**

This paper presents ToaSt, a compression method designed to make ViTs more efficient. After rebuttal, it received scores of 2544. Most reviewers agree that the paper is well written, and results are comprehensive. The proposed method is well-explained with clear illustration. And most concerns have been addressed after rebuttal except weakness 11 proposed by reviewer advx. The AC agreed with this, and thinks the author should further verify the accuracy of the data in Table 5.

Overall, the AC thinks the merits outweigh the flaws, therefore, would like to recommend acceptance of the paper.